# Multiplexed imaging of immune cells in staged multiple sclerosis lesions by mass cytometry

Valeria Ramaglia[1]*, Salma Sheikh-Mohamed[1], Karen Legg[1], Calvin Park[2], Olga L Rojas[1], Stephanie Zandee[3], Fred Fu[4], Olga Ornatsky[5], Eric C Swanson[5], David Pitt[2], Alexandre Prat[3], Trevor D McKee[4], Jennifer L Gommerman[1]

[1]Department of Immunology, University of Toronto, Toronto, Canada; [2]Department of Neurology, Yale School of Medicine, New Haven, United States; [3]Department of Neuroscience, Faculty of Medicine, Université de Montréal, Montreal, Canada; [4]STTARR Innovation Centre, University Health Network, Toronto, Canada; [5]Fluidigm Canada Inc, Markham, Canada

**Abstract** Multiple sclerosis (MS) is characterized by demyelinated and inflammatory lesions in the brain and spinal cord that are highly variable in terms of cellular content. Here, we used imaging mass cytometry (IMC) to enable the simultaneous imaging of 15+ proteins within staged MS lesions. To test the potential for IMC to discriminate between different types of lesions, we selected a case with severe rebound MS disease activity after natalizumab cessation. With post-acquisition analysis pipelines we were able to: (1) Discriminate demyelinating macrophages from the resident microglial pool; (2) Determine which types of lymphocytes reside closest to blood vessels; (3) Identify multiple subsets of T and B cells, and (4) Ascertain dynamics of T cell phenotypes vis-à-vis lesion type and location. We propose that IMC will enable a comprehensive analysis of single-cell phenotypes, their functional states and cell-cell interactions in relation to lesion morphometry and demyelinating activity in MS patients.
DOI: https://doi.org/10.7554/eLife.48051.001

*For correspondence:
v.ramaglia@utoronto.ca

## Introduction

Multiple sclerosis (MS) is a disease with profound heterogeneity in the neuropathological and immunopathological appearance of lesions in the central nervous system (CNS) (*Lucchinetti et al., 2000*). Recent consensus has standardized staging of MS brain tissue into categories including normal-appearing white matter (NAWM), (p)reactive lesions (or 'pre-phagocytic' lesions) (*Barnett and Prineas, 2004*) which may represent an initial lesion (*Marik et al., 2007*; *Alvarez et al., 2015*), periplaque white matter (PPWM) which is immediately adjacent to a lesion, early or late active demyelinating lesions, mixed active/inactive demyelinating lesions (also called slowly expanding or 'smouldering'; see *Frischer et al., 2015*), and inactive lesions (*Kuhlmann et al., 2017*). The pattern of demyelination can also be fundamentally different between patients, with pattern I being T cell-mediated, pattern II being IgG- and complement-mediated, and pattern III and IV characterized by a primary oligodendrocyte dystrophy reminiscent of virus- or toxin-induced demyelination rather than autoimmunity (*Lucchinetti et al., 2000*).

Lymphocytes, microglia and macrophages are associated with active demyelination and neurodegeneration in the MS brain (*Frischer et al., 2009*; *Haider et al., 2014*) and are thought to play key roles in the disease process, as supported by studies in experimental models (reviewed in *Robinson et al., 2014*). Depending on the type of lesion and the sub-region within a lesion (for example center vs edge), different myeloid and lymphoid cells can be found. These have a variety of

**eLife digest** It takes an army of immune cells to defend the body against infection. But sometimes the body's immune system mistakenly attacks its own cells and chronic inflammatory conditions develop. In multiple sclerosis – also known as "MS" – a horde of immune cells infiltrate the brain and spinal cord, forming lesions which strip nerve cells of their insultation, a protective fatty material called myelin. Nerve cells become damaged, scarred and exposed, and this interferes with messages between the brain and other parts of the body.

Advanced imaging techniques have revolutionized the diagnosis of multiple sclerosis by capturing lesions as they develop in the brain and spinal cord. Researchers have also focused their efforts on understanding how immune cells activated in the blood stream invade the central nervous system. To better understand how a mistaken immune response leads to nerve damage in multiple sclerosis, a forensic examination of which immune cells accumulate in brain tissue to form lesions is needed. Standard techniques for analyzing whole tissue samples are however limited by design, capable of detecting only a few cell markers in one section of tissue.

Ramaglia et al. have now validated a new imaging technique for looking at an array of cell types in brain tissue in a single sample. The technique – called imaging mass cytometry (or IMC for short) – was used to look at post-mortem brain tissue from a multiple sclerosis patient with an acute form of the illness. The tissue examined had multiple sclerosis lesions present. Different types of immune cells were simultaneously identified and characterized using a panel of antibodies which recognize the signature proteins each immune cell makes when active. The state of the underlying myelin content of the tissue was also characterized.

The imaging approach could distinguish between the immune cells of the brain (known as resident microglia) and a type of white blood cell summoned as part of the immune response (infiltrating macrophages). The analysis showed that, in the particular patient examined, microglia are abundant in active lesions in multiple sclerosis; also, different subsets of white blood cells were detected. Measuring how far different immune cells had migrated from nearby blood vessels added insights as to how immune cells move through the brain and which cells may have arrived first.

Altogether, Ramaglia et al. have shown that IMC can be used as a discovery tool to gain a deeper understanding of multiple sclerosis lesions and immune cells active in the inflamed brain. Further work will apply this now validated imaging approach to large cohorts of multiple sclerosis patients.

DOI: https://doi.org/10.7554/eLife.48051.002

phenotypes that reflect activation state and pathologic potential. With respect to myeloid cells, yolk sac-derived (resident) microglia and blood-derived (recruited) monocytes/macrophages accumulate at sites of active demyelination and neuroaxonal injury (*Frischer et al., 2009*). Microglia and macrophages within the MS brain can lose their normally homeostatic properties and acquire a pro-inflammatory phenotype with expression of molecules involved in phagocytosis, oxidative injury, antigen presentation and T cell co-stimulation (*Gay et al., 1997*). Either via T cell-mediated recognition of myelin epitopes (*Sun et al., 2001*) or complement binding to myelin autoantibodies (*Storch et al., 1998*), these lymphocyte-dependent events initiate a process that results in the activation of microglia and recruitment of macrophages at the lesion site. Microglia and macrophages become activated and internalize myelin, degrading it within their lysosomes. The detection of small (myelin oligodendrocyte glycoprotein, MOG) or large (proteolipid protein, PLP) myelin proteins indicates the temporal development of myelin destruction (*Lucchinetti et al., 2000*; *Kuhlmann et al., 2017*).

In terms of lymphocytes, MS lesions contain T cells and CD20$^+$ B cells (*Machado-Santos et al., 2018*). In active lesions, CD8$^+$ T cells proliferate and have an activated cytotoxic phenotype. Subsequently, some CD8$^+$ T cells are destroyed by apoptosis while others, with tissue-resident memory features, persist. Tissue resident memory T cells lose expression of surface molecules that are involved in the egress of leukocytes from inflamed tissue, which has been suggested as a potential mechanism responsible for the compartmentalized inflammatory response in established lesions (*Machado-Santos et al., 2018*). CD4$^+$ T cells are also found in MS lesions and have been shown to produce cytokines such as IL-17 and IFNγ (*Kebir et al., 2007*). B cells are thought to differentiate

into plasma cells, perhaps in situ, but little is known about their phenotype (*Machado-Santos et al., 2018*).

While past and recent immunohistological studies have provided insights into the types of immune cells populating MS lesions at different lesional stages and the neurodegenerative changes that accompany these infiltrating immune cells (*Frischer et al., 2009*; *Machado-Santos et al., 2018*; *Luchetti et al., 2018*; *Zrzavy et al., 2017*; *Popescu et al., 2017*; *Fischer et al., 2012*), this type of analysis requires immunohistological staining of serial sections and is limited to the number of analytes that can be simultaneously visualized on a given tissue section. Thus, a comprehensive analysis of single-cell phenotypes and functional states in relation to demyelination within MS tissue is lacking. To circumvent this challenge, we have employed imaging mass cytometry (IMC). IMC uses time-of-flight inductively coupled plasma mass spectrometry to detect dozens of markers simultaneously on a single tissue section. It achieves this by measuring the abundance of metal isotopes tagged to antibodies and indexed against their source location (*Chang et al., 2017*). Applying this new technology to post-mortem MS brain tissue, we carefully analysed staged lesions in a case with severe rebound MS disease activity after natalizumab (NTZ, also known as anti-VLA4) cessation (*Larochelle et al., 2017*). The highly inflammatory nature of the lesions present in the brain of this specific case, makes this tissue a good positive control to test a variety of analytes involved in neuroinflammation. The data we collected suggest that IMC, in combination with existing imaging techniques, can profoundly impact our knowledge of the inflammatory response and tissue injury in the MS brain.

## Results

### Comparability of IF versus IMC approach and specificity of metal-conjugated antibodies on brain-resident cell types

In this study, we performed two separate IMC runs on the same patient and control tissue. For the patient, we selected brain tissue from a NTZ-rebound case with significant neuroinflammation and multiple types of lesions apparent including (p)reactive, active, mixed active-inactive as well as normal appearing white matter. The first run (*Figures 1–2*) was to validate our panel and the approach in general. The second run (*Figures 3–7*) was to evaluate the composition of immune cells across several types of lesions and sub-lesional areas. To evaluate the validity of IMC for the analysis of post-mortem MS brain tissue, we first investigated whether images generated by IMC revealed a similar number of cells expressing a given marker in a mm$^2$ (*Barnett and Prineas, 2004*) area of tissue, as determined by IF. We used IF or IMC on serial sections from the same tissue block. Sections were stained with DAPI (IF) or Iridium-intercalator (IMC) to identify cell-associated DNA, and with anti-CD3-FITC (IF) or anti-CD3-170Er (IMC) to identify CD3$^+$ T cells. Imaging of equivalent ROI in IF- and IMC-stained sections showed similar staining patterns with clearly resolved anatomical regions (*Figure 1a*). We also showed that IMC is able to resolve myelin engulfed by microglia/macrophages with a similar pattern as what is observed using IHC and IF approaches (*Figure 1—figure supplement 3*). Finally, we performed a number of comparisons of analytes imaged using IF versus IMC. Comparisons included (1) the number of nuclei identified with DAPI by IF and the number of nuclei identified with intercalator by IMC (*Figure 1b*); (2) the number of CD3$^+$ T cells identified with FITC-labeled secondary antibody by IF and the number of CD3$^+$ T cells identified with the 170Er metal-tag by IMC (*Figure 1c*); (3) the percentage of PLP$^+$ area identified with AF568-labeled secondary antibody by IF and the percentage of PLP$^+$ area identified with the 141Pr metal-tag by IMC (*Figure 1—figure supplement 4a*); (4) the number of HLA$^+$ myeloid cells identified with FITC-labeled secondary antibody by IF and the number of HLA$^+$ myeloid cells identified with the 147Sm metal-tag by IMC (*Figure 1—figure supplement 4b*); and (5) the number of CD68$^+$ myeloid cells identified with FITC-labeled secondary antibody by IF and the number of CD68$^+$ myeloid cells identified with the 159Tb metal-tag by IMC (*Figure 1—figure supplement 4c*). All comparisons yielded a significant positive correlation (Spearman correlation coefficient: r = 0.9182, p=0.0002; r = 0.8929, p=0.01; r = 0.9544, p<0.0001; r = 0.9794, p<0.0001; r = 0.9051, p<0.0001 for nuclei, CD3$^+$ T cells, %PLP$^+$ area, HLA$^+$ cells and CD68$^+$ cells respectively), indicating proportional representation of brain-resident cells that is in agreement for both methods. In summary, IMC reproduces staining patterns that are in agreement with those produced using the more standard IF method in an NTZ-rebound brain tissue.

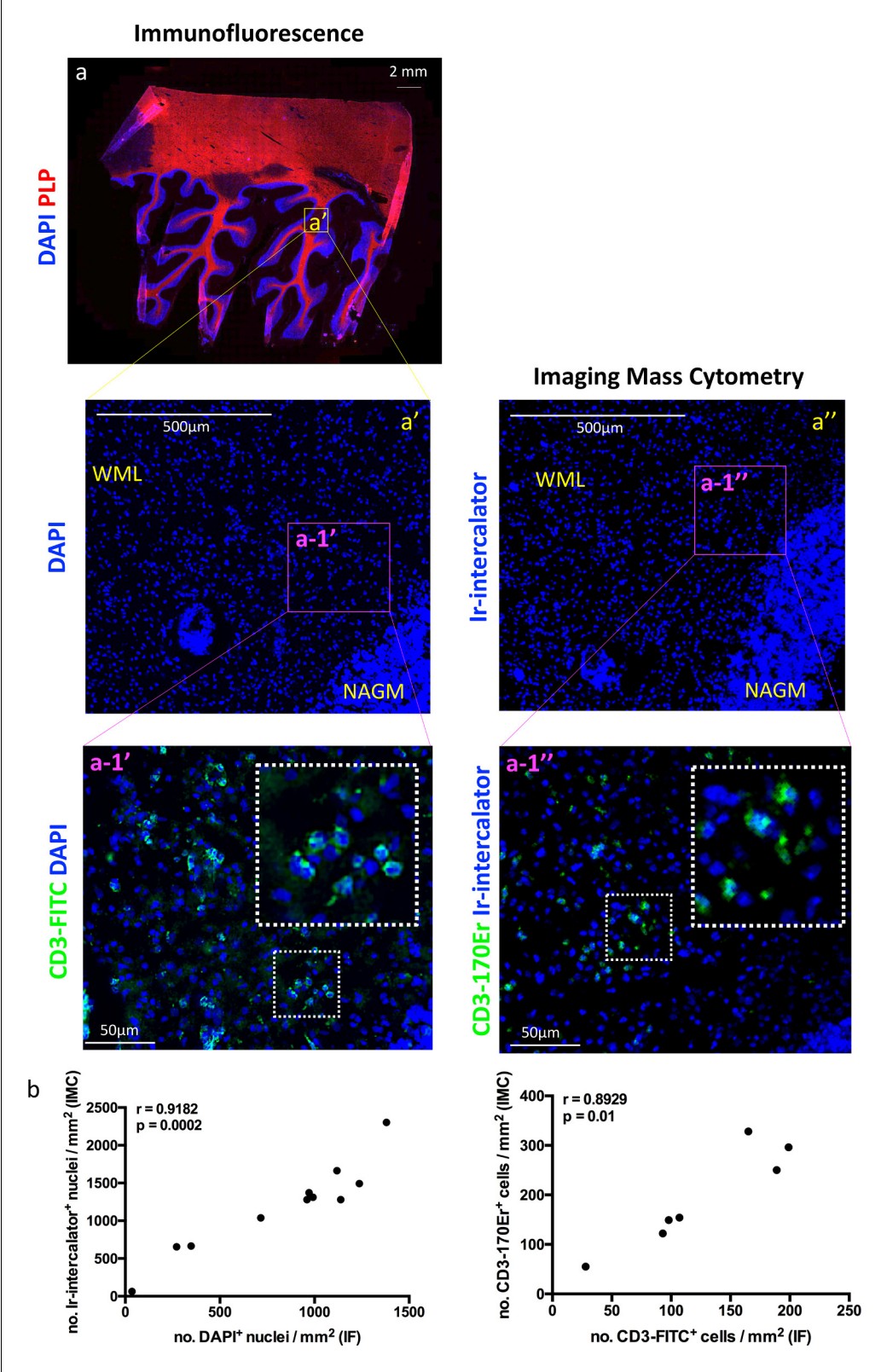

**Figure 1.** Comparing IMC to IF in MS lesions. Two serial sections were assessed: one used for immunofluorescence (IF, **a and a'**) and one dedicated to Imaging Mass Cytometry (IMC, **a''**). (a) The region of interest (**a'**) was guided by the immunofluorescence staining with anti-PLP (proteolipid protein, shown in red to visualize myelin), and DAPI (shown in blue to visualize nuclei) for the identification of lesion location and type (see *Figure 1—figure supplement 1*). The entire region of interest on a serial section was subjected to IMC, according to the work flow shown in *Figure 1—figure*

*Figure 1 continued on next page*

*Figure 1 continued*

**supplement 2**. Staining with Iridium (Ir)-intercalator is shown in blue to visualize DNA in nuclei. A blow up area of the region of interest within an active lesion (referred to as a-1′ for IF and a-1′′ for IMC), was also stained with fluorochrome conjugated anti-CD3 (a-1′) or metal conjugated anti-CD3 (a-1′), both depicted in green. WML, white matter lesion; NAGM, normal-appearing gray matter. (**b**) Spearman correlation coefficient, showing a significant positive correlation between the number of nuclei identified with DAPI by IF and the number of nuclei identified with Ir-intercalator by IMC (n = 11, coefficient, r = 0.9182, p=0.0002). (**c**) Spearman correlation coefficient, showing a significant positive correlation between the number of CD3$^+$ T cells identified with fluorochrome-conjugated antibody by IF and the number of CD3$^+$ T cells identified with metal-conjugated antibody by IMC (n = 7, coefficient, r = 0.8929, p=0.01). Additional correlation analyses between fluorochrome-conjugated antibodies by IF and metal-conjugated antibodies by IMC are shown in *Figure 1—figure supplement 4*.

DOI: https://doi.org/10.7554/eLife.48051.003

The following figure supplements are available for figure 1:

**Figure supplement 1.** Staging of MS lesions by IF.

DOI: https://doi.org/10.7554/eLife.48051.004

**Figure supplement 2.** Workflow of Imaging Mass Cytometry.

DOI: https://doi.org/10.7554/eLife.48051.005

**Figure supplement 3.** Validation of IMC staining patterns in MS lesions.

DOI: https://doi.org/10.7554/eLife.48051.006

**Figure supplement 4.** Correlating IMC to IF staining patterns in MS lesions.

DOI: https://doi.org/10.7554/eLife.48051.007

Second we assessed the target specificity of metal-tagged antibodies by IMC by staining for proteins that are either expected to be co-expressed by cells, or whose cellular expression is expected to be mutually exclusive. While each region of interest is stained with the entire antibody panel simultaneously, and the identification of cell phenotypes is based on all markers included in the panel (using negative and positive selection as described in the Materials and methods), we limited our depiction of the images to display no more than three analytes at once. An example of an image with 8-analytes displayed simultaneously is depicted in *Figure 2—figure supplement 1*. IMC imaging of the edge of an active demyelinating lesion identified CD3$^+$CD45$^+$ T cells and CD3$^-$CD45$^+$ leukocytes other than T cells (*Figure 2a*). The co-expression of CD68 in CD3$^-$CD45$^+$ cells identifies these as microglia/macrophages (CD3$^-$CD45$^+$CD68$^+$) (*Figure 2b*). Since antibodies directed to the B cell-restricted lineage markers CD19 and CD20 were sub-optimal on our brain tissues, we relied instead on antibodies that detect the two allelic variants of the immunoglobulin light chain (κ/λ). This provided two advantages: the ability to capture all B cells, irrespective of their maturation/activation status (for example, plasma cells downregulate CD19/CD20) and the ability to test specificity of our B cell directed reagents since κ and λ are allelically excluded on the surface of B cells. As expected, CD3$^+$ T cells lack expression of immunoglobulin light chain, (CD3$^+$κ$^-$/λ$^-$). Moreover, we identified B cells that were either positive for the κ or λ light chain and negative for the CD3 marker of T cells and the CD68 marker of macrophages (for example, CD3$^-$CD68$^-$κ$^+$ B cells in *Figure 2c* arrow, versus CD3$^-$CD68$^-$λ$^-$ B cells in *Figure 2d* arrow and vice versa CD3$^-$CD68$^-$κ$^-$ B cells in *Figure 2c* arrow head, versus CD3$^-$CD68$^-$λ$^+$ B cells in *Figure 2d* arrow head). We found that the extracellular matrix protein collagen (not cell-associated) surrounded putative blood vessels lined by endothelial cells that expressed the CD31 marker as expected. In contrast, macrophages visualized by CD68 do not stain positive for collagen nor express CD31 (CD31$^-$collagen$^-$CD68$^+$) (*Figure 2e*). Lastly, we asked whether the IMC approach would have sufficient sensitivity to detect soluble molecules that can be rare in tissues. For this, we used an antibody against granzyme B, a serine protease with pro-inflammatory function produced by activated cytotoxic T cells. IMC identified a granzyme B$^+$ signal with granular expression in close proximity to nuclei (*Figure 2f,g*). These results show that IMC enables imaging of multiple markers on a single tissue section reproducing IF-equivalent staining patterns and with cell lineage-specific markers expressed on appropriate cell types.

## Qualitative staging of NTZ rebound lesions by IMC

To verify whether IMC provides us with the ability to differentiate normal-appearing tissue versus different lesional stages of the NTZ rebound brain, we analysed the proteolipid protein (PLP) signal that visualizes myelin and the human leukocyte antigen (HLA) signal that visualizes antigen presenting cells (*Figure 3*). Note that in some ROI, the PLP staining pattern reflects the cross-sectional

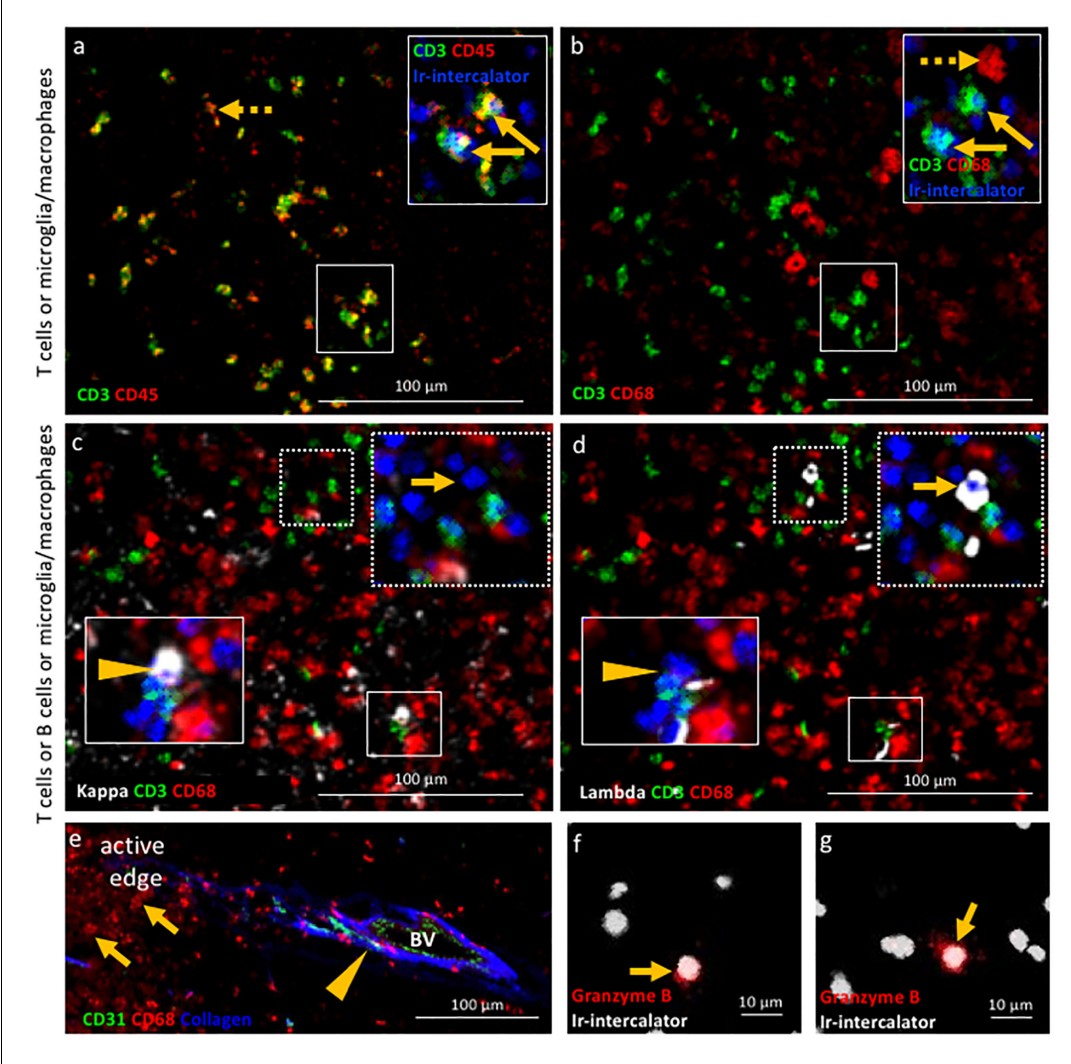

**Figure 2.** Validation of IMC specificity in MS lesions. (a) Overlay of CD3 (green) and CD45 (red) identifies CD3+CD45+ T cells (solid arrows) and CD3-CD45+ leukocytes other than T cells (dotted arrow). (b) Overlay of CD3 (green) and CD68 (red) identifies CD3+CD68- T cells (solid arrows) and CD3-CD68+microglia/macrophages (dotted arrow). Note that the solid arrows in a and b indicates the same CD3+CD45+CD68- T cells. (c) Overlay of κ (white), CD3 (green) and CD68 (red) and (d) overlay of λ (white), CD3 (green) and CD68 (red) identify κ+CD3-CD68- B cells (arrow head in c) that are λ-CD3-CD68- (arrow head in d) and κ-CD3-CD68- B cells (arrow in c) that are λ+CD3-CD68- (arrow in d), as expected based on the allelic exclusion of κ and λ. (e) Overlay of CD31 (green), CD68 (red) and Collagen (blue) identifies CD31+Collagen+CD68- endothelial cells (arrow head) and CD31-Collagen-CD68+ microglia/macrophages (arrows). (f, g) Granzyme B+ cells (arrows). Images in (a and b) as well as images in c) and d) are from the same areas of an active demyelinating lesion. Image in (e) are from the edge of an active demyelinating lesion. Images in (f and g) are from the center of an active demyelinating lesion.

DOI: https://doi.org/10.7554/eLife.48051.008

The following figure supplement is available for figure 2:

**Figure supplement 1.** Example of active lesion in which 8 analytes are displayed simultaneously.
DOI: https://doi.org/10.7554/eLife.48051.009

orientation of myelinated fibers (for example WMC and NAWM), whereas in others, longitudinal myelin tracks are observed (for example (p)reactive). The different orientation of the tissue results from the sectioning plane of the tissue block and is reflected in the staining pattern displayed. Consistent with the generally non-inflamed and myelinated state of healthy white matter, control white matter showed intact myelin staining with few HLA+ cells (*Figure 3a,b*). Normal appearing white matter (NAWM) in the NTZ rebound brain exhibited normal myelin staining, however HLA+ cells within the NTZ rebound NAWM appeared enriched when compared to control white matter

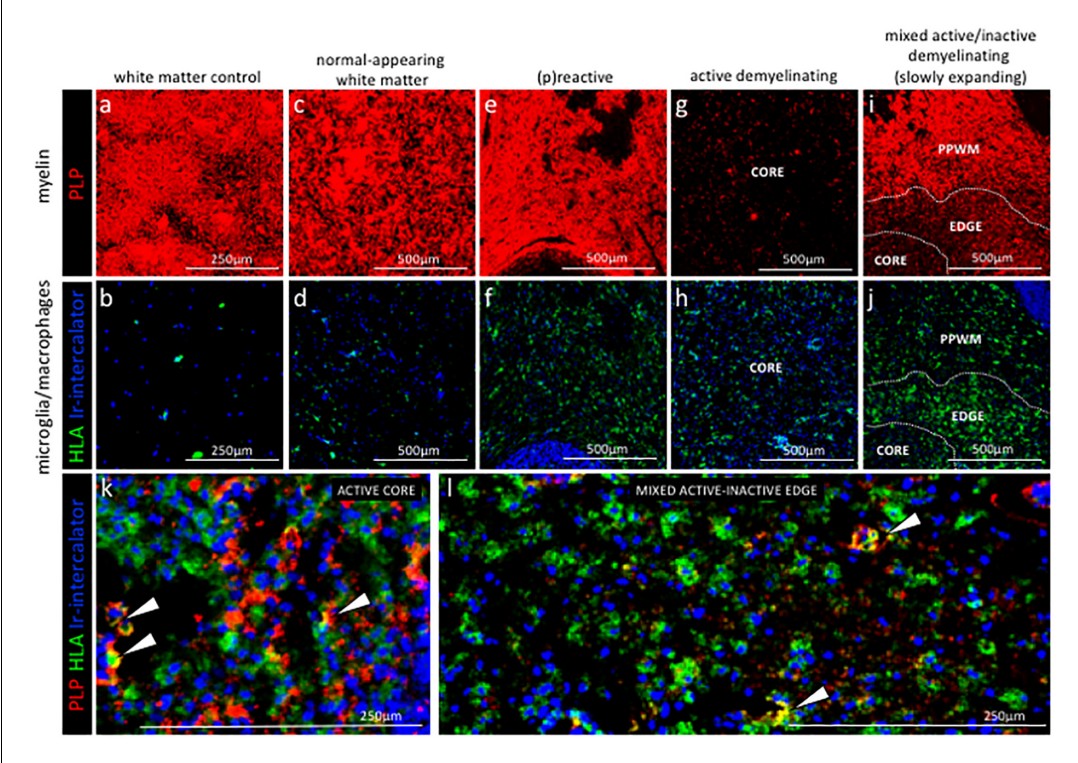

**Figure 3.** Staging of MS lesions by IMC. Representative mass cytometry images of white matter areas of (**a, f**) healthy control, (**b, g**) MS normal-appearing white matter (block no. CR4A), (**c, h**) MS (p)reactive lesion (block no. CR4A), (**d–i**) MS active demyelinating lesion (block no. CR4A) and (**e–j**) an MS mixed active-inactive demyelinating lesion (block no. CL3A). For each region of interest, we show the same area simultaneously labeled with markers of myelin (proteolipid protein, PLP), antigen presentation (human leukocyte antigen, HLA) to detect microglia/macrophages and DNA (intercalator). (**a–e**) Images of PLP (red) and (**f–i**) overlay of HLA (green) and intercalator (blue) show the lesion activity in staged MS lesions compared to control white matter and normal-appearing white matter. (**k, l**) Overlay of PLP, HLA and intercalator show microglia/macrophages containing PLP[+] myelin protein in the core of (**k**) an active lesion and (**l**) in the edge of a slowly expanding lesion, indicative of demyelinating activity. PPWM, periplaque white matter; BV, blood vessel.

DOI: https://doi.org/10.7554/eLife.48051.010

(*Figure 3c,d*). Similarly, the (p)reactive lesion showed a normal myelin signal but HLA[+] cells accumulated at this site (*Figure 3e,f*). The active lesion core showed loss of PLP signal with accumulation of HLA[+] cells (*Figure 3g,h*). The mixed active-inactive lesions showed reduced myelin and accumulation of HLA[+] cells at the lesion edge (*Figure 3i,j*). HLA[+] cells that contained PLP myelin products were found in both active lesions (*Larochelle et al., 2017*) (*Figure 3k*) and at the edge of mixed active-inactive lesions (*Figure 3l*), indicative of demyelinating activity. Collectively we were able to show that application of IMC to different ROI (pre-selected on the bases of PLP/HLA IF staining on a serial section) was able to differentiate between normal-appearing tissue and different lesional stages of the NTZ rebound brain.

## Qualitative assessment of microglia and macrophages in staged NTZ-rebound lesions by IMC

Next, we analysed key molecules that differentiate between the phenotype and functional status of microglia and macrophages in relation to the lesional stage and demyelinating activity of NTZ rebound lesions.

*Control subject white matter.* In the white matter from a control subject we found that microglia, identified as being TMEM119[+21], generally showed a thin ramified morphology, typical of resting cells (*Figure 4a,a'* dotted arrows). On these cells, the HLA marker of antigen presentation was generally low or not detectable, confirming a quiescent state. On the contrary, TMEM119[+] microglia that showed a more rounded morphology, a sign of activation, also stained positive for HLA and

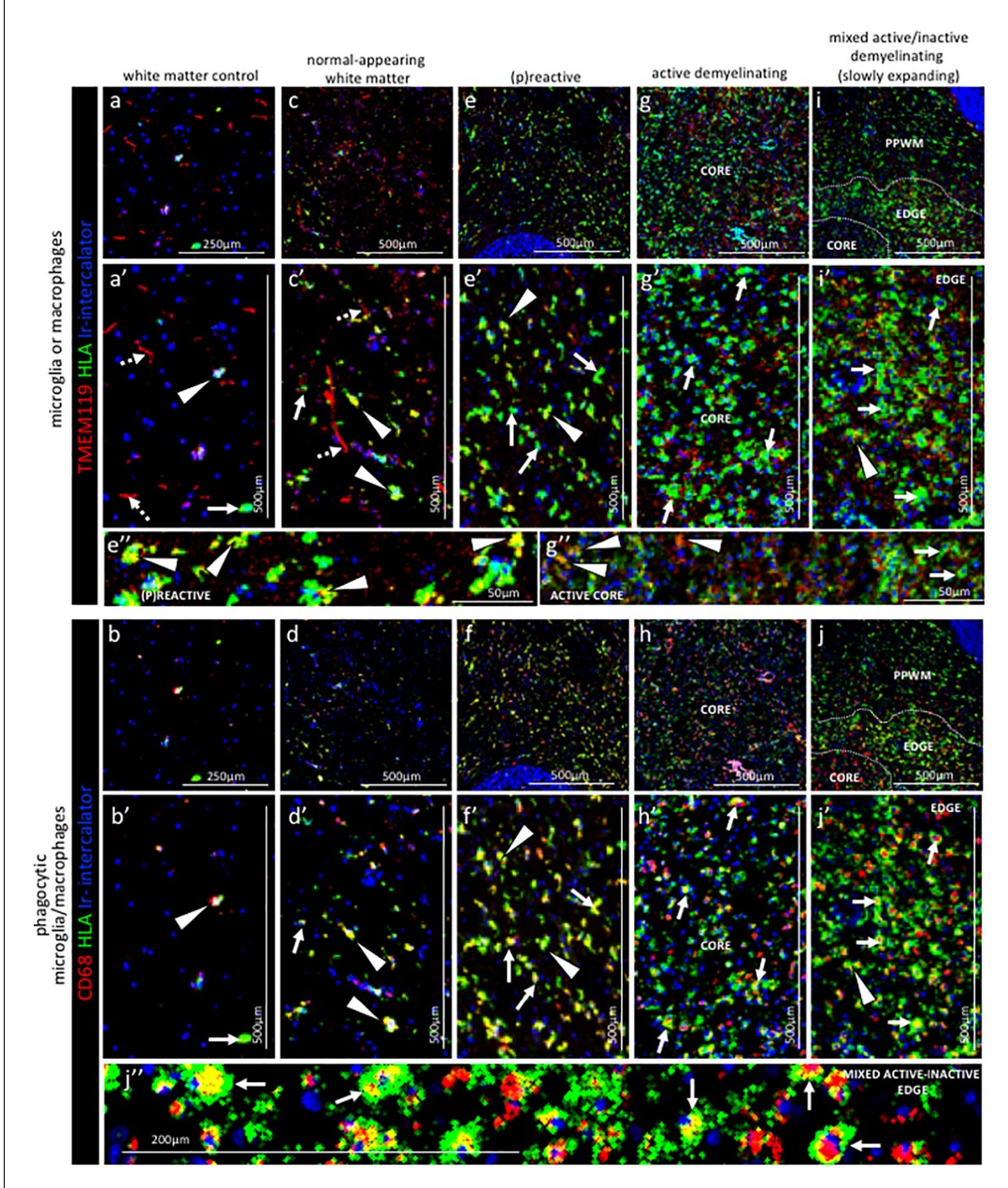

**Figure 4.** Pattern of microglia or macrophage activity in different stages of MS lesions by IMC. Representative mass cytometry images of (a, a', b, b') control white matter, (c, c', d, d') normal-appearing white matter (block no. CR4A), (e, e', f, f') (p)reactive lesion (block no. CR4A), (g, g', h, h') active demyelinating lesion (block no. CR4A) and (i, i', j, j') mixed active-inactive demyelinating lesion (block no. CL3A). For each region of interest, we show the same area simultaneously labeled with markers of antigen presentation (human leukocyte antigen, HLA) to detect microglia and/or macrophages, TMEM119 to detect microglia, lysosomes (CD68) to detect phagocytic cells and DNA (Ir-intercalator). (a, a'– i, i') Overlay of TMEM119 (red), HLA (green) and Ir-intercalator (blue) identifies (dotted arrows in a' and c') TMEM119$^+$HLA$^-$ resting microglia with thin elongated processes and (arrows head in a',c', e', i' and e'') TMEM119$^+$HLA$^+$ activated microglia or (solid arrows in a', c', e', g', i' and g'') TMEM119$^-$HLA$^+$ activated macrophages. (b, b'–j, j'') Overlay of CD68 (red), HLA (green) and Ir-intercalator (blue) identifies HLA$^+$CD68$^+$ phagocytic microglia/macrophages. PPWM, periplaque white matter; BV, blood vessel.

DOI: https://doi.org/10.7554/eLife.48051.011

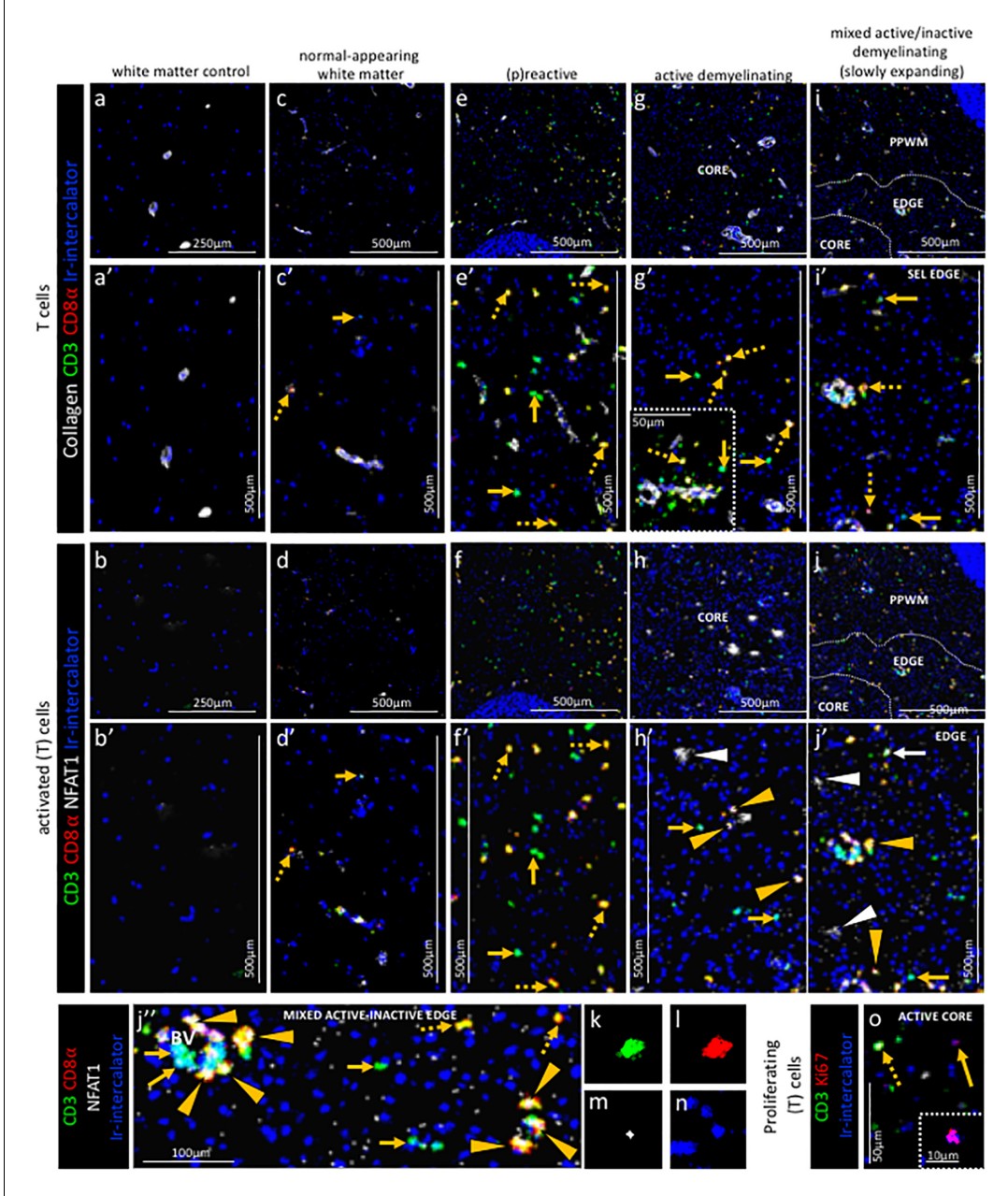

**Figure 5.** Pattern of T cell subpopulations in different stages of MS lesions by IMC. Representative mass cytometry images of (**a, a', b, b'**) white matter of control, (**c, c', d, d'**) normal-appearing white matter (block no. CR4A), (**e, e', f, f'**) (p)reactive lesion (block no. CR4A), (**g, g', h, h', o**) active demyelinating lesion (block no. CR4A) and (**i, i', j–n**) mixed active-inactive demyelinating lesion (block no. CL3A). For each region of interest, we show the same area simultaneously labeled with anti-collagen antibodies to visuzliae blood vessels, all T cells (CD3), CD8α T cells, cell proliferation (Ki67) and DNA (Ir-intercalator). (**a, a'– i, i'**) Overlay of collagen (white), CD3 (green), CD8α (red) and Ir-intercalator (blue) identifies (dotted arrows in **c', e', g'** and **i'**) CD3$^+$CD8α$^+$ T cells, (solid arrows in **c', e', g'** and **i'**) CD3$^+$CD8α$^-$ (therefore by exclusion putative CD4$^+$) T cells and collagen$^+$ blood vessels. (**b–b'–j, j''**) Overlay of CD3 (in green), CD8α (red), NFAT1 (in white) and Ir-intercalator (in blue) identifies (yellow arrow head in **h', j'** and **j''**) CD3$^+$CD8α$^+$NFAT1$^+$ T cells and (white solid arrow in **j'**) CD3$^+$CD8α$^-$NFAT1$^+$ (putative CD4$^+$) T cells. (white arrow head in **h'** and **j'**) CD3$^-$CD8α$^-$NFAT1$^+$ cells are also detected. (**o**) Overlay of CD3 (in green), Ki67 (red) and Ir-intercalator (in blue) identifies CD3$^+$Ki67$^+$ proliferating T cells (dotted arrow) and CD3$^-$Ki67$^+$ proliferating cells other than T cells (solid arrows and inset). PPWM, periplaque white matter.

DOI: https://doi.org/10.7554/eLife.48051.012

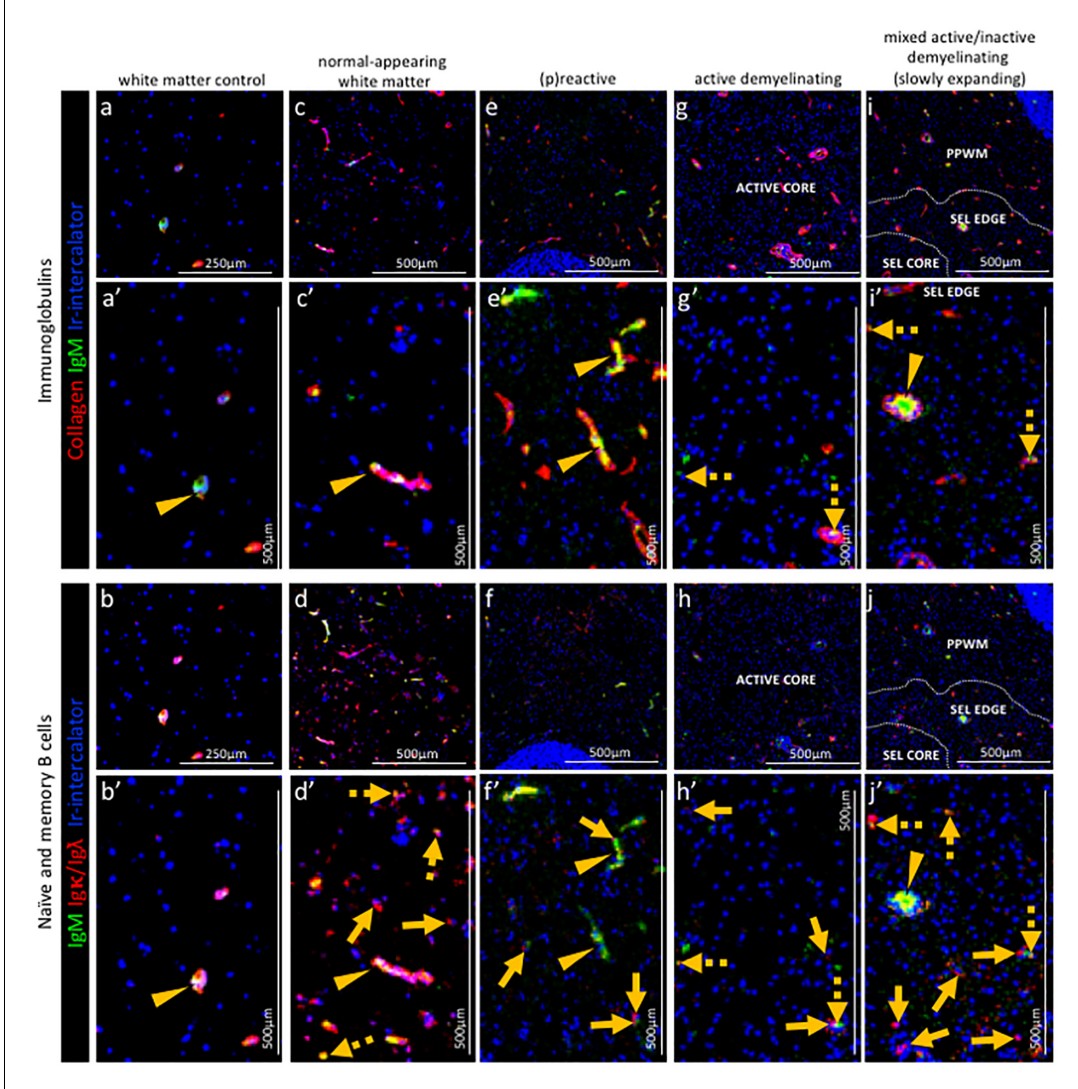

**Figure 6.** Pattern of immunoglobulins and B cell subpopulations in different stages of MS lesions by IMC. Representative mass cytometry images of (**a, a', b, b'**) white matter of control, (**c, c', d, d'**) normal-appearing white matter (block no. CR4A), (**e, e', f, f'**) a (p)reactive lesion (block no. CR4A), (**g, g', h, h', o**) an active demyelinating lesion (block no. CR4A) and (**i, i', j–n**) a mixed active-inactive demyelinating lesion (block no. CL3A). For each region of interest, we show the same area simultaneously labeled with markers of endothelial cells (collagen) to detect blood vessels, immunoglobulin M (IgM), the κ or λ light chain of immunoglobulins (Igκ/Igλ) to detect B cells and DNA (Ir-intercalator). (**a, a–i, i'**) Overlay of collagen (red), IgM (green) and Ir-intercalator (blue) identifies cellular (intercalator-associated, dotted arrows in **g'** and **i'**) and non-cellular (free immunoglobulin, arrows head in **a', c', e', i'**) IgM in the parenchyma or within collagen⁺ blood vessels. (**b, b'–j, j'**) Overlay of IgM (green), Igκ/Igλ (red) and Ir-intercalator (blue) identifies (dotted arrow in **d', h' and j'**) Igκ/Igλ⁺IgM⁺ naïve and IgM memory B cells and (solid arrows in **d', f', h' and j'**) Igκ/Igλ⁺IgM⁻ class switch B cells.

DOI: https://doi.org/10.7554/eLife.48051.013

CD68 which is indicative of antigen presentation and phagocytic activity, respectively (**Figure 4a,a'** arrow head and b, b' arrow head). TMEM119⁻HLA⁺CD68⁺ cells were identified as macrophages and were also present in the white matter of a control (**Figure 4a,a'** arrow and b, b' arrow). These data indicate that in the normal white matter of a control subject some microglia (TMEM119⁺) and some macrophages (TMEM119⁻) have an activated phenotype (HLA⁺CD68⁺).

*Normal-appearing white matter.* Visualization of the expression pattern of microglia and macrophage markers in the normal-appearing white matter showed some TMEM119⁺ microglia with ramified morphology (**Figure 4c,c'** dotted arrows), similar to control white matter. However unlike

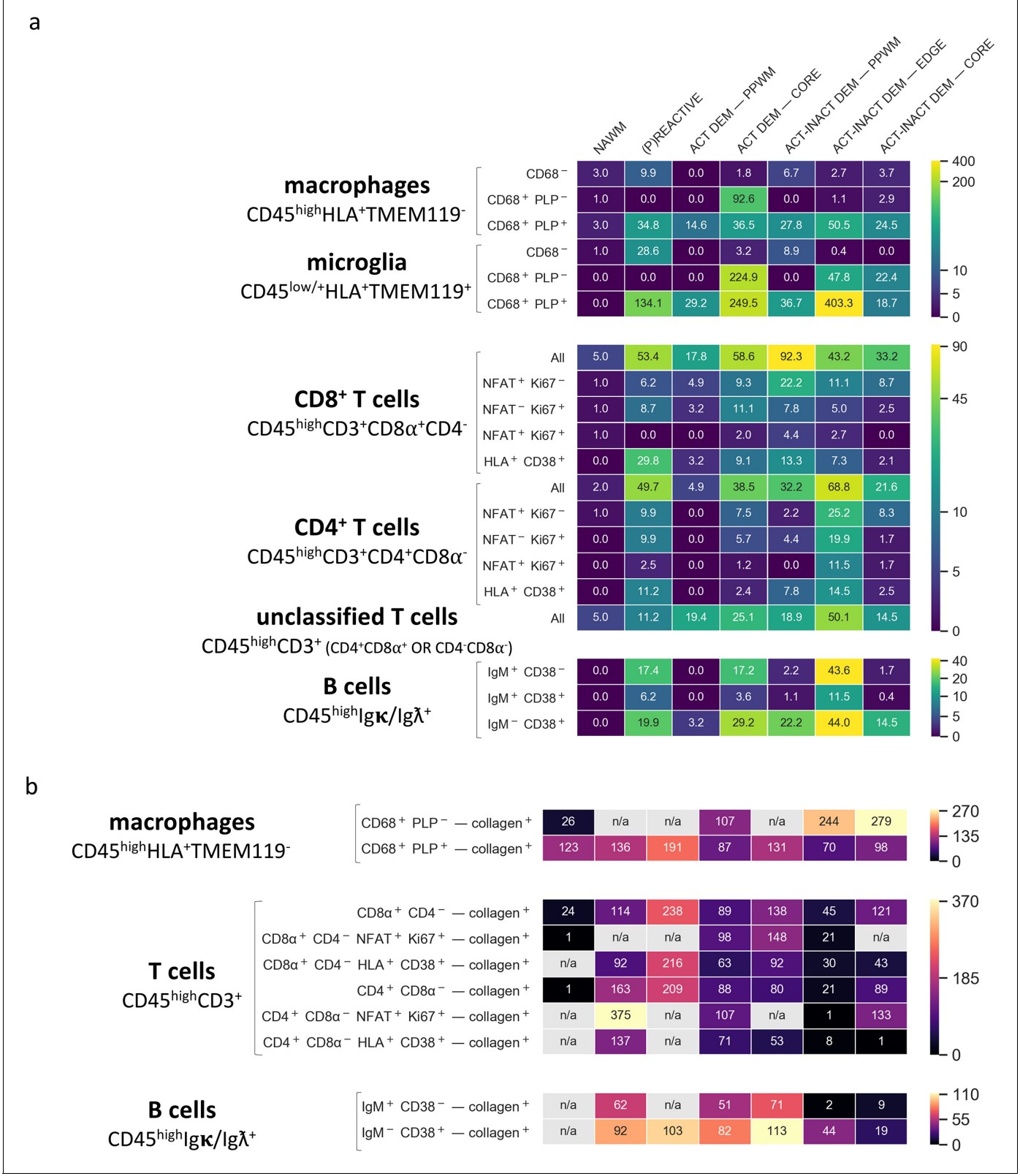

**Figure 7.** Density of immune cell subsets in different stages of MS lesions and their distance from blood vessels by IMC. (**a**) Cell counts are provided as number of cells per mm$^2$ (*Barnett and Prineas, 2004*) of region of interest. The category of cells is defined according to the expression of cell-specific and functional markers as indicated and also described in *Table 2*. (**b**) Distance between defined categories of cells and blood vessels (collagen$^+$) are provided in µm. NAWM, normal-appearing white matter; PPWM, periplaque white matter; Act dem, active demyelinating; act inact dem, active-inactive

*Figure 7 continued on next page*

*Figure 7 continued*

demyelinating. The single-cell segmentation strategy is shown in *Figure 7—figure supplement 1*. The Positive and negative 'gates' used to identify each cell subset were established based on the quadrants defined by manually-identified cells according to the pipeline shown in *Figure 7—figure supplements 2–4* and laid out in *Figure 7—figure supplement 5*. Please see the section 'Gating strategy for quantitative analysis of T cell, B cell, macrophage and microglial cell subsets' in the Materials and methods. The gating strategy used for the generation of heat maps is laid out in *Figure 7—figure supplement 6*. Source files used for the quantitative analysis are provided in *Figure 7—source data 1*.
DOI: https://doi.org/10.7554/eLife.48051.014

The following source data and figure supplements are available for figure 7:

**Source data 1.** Source file for quantitative data of all ROI.
DOI: https://doi.org/10.7554/eLife.48051.021
**Figure supplement 1.** Single cell segmentation and validation of approach using anti-CD3.
DOI: https://doi.org/10.7554/eLife.48051.015
**Figure supplement 2.** Manual selection of myeloid cells.
DOI: https://doi.org/10.7554/eLife.48051.016
**Figure supplement 3.** Manual selection of T cells.
DOI: https://doi.org/10.7554/eLife.48051.017
**Figure supplement 4.** Manual selection of B cells.
DOI: https://doi.org/10.7554/eLife.48051.018
**Figure supplement 5.** Gating strategy used for the identification of cell subsets.
DOI: https://doi.org/10.7554/eLife.48051.019
**Figure supplement 6.** Gating strategy used for the generation of heat maps.
DOI: https://doi.org/10.7554/eLife.48051.020

control white matter, the normal-appearing white matter showed many TMEM119$^+$ microglia that were also positive for HLA and CD68 (*Figure 4c,c'* arrows head and d, d' arrows head). A few TMEM119$^-$HLA$^+$CD68$^+$ macrophages were also present in the normal-appearing tissue (*Figure 4c,c'* arrow and d, d' arrow).

(P)*reactive lesions.* Within the (p)reactive lesions, TMEM119$^+$ microglia accumulated, showed an enlarged morphology that is indicative of an activated state, and expressed both HLA and CD68 (*Figure 4e,e' and e''* arrows head and f, f' arrows head). TMEM119$^-$HLA$^+$CD68$^+$ macrophages were also present (*Figure 4e,e'* arrows and f, f' arrows).

*Active lesions.* Active lesions contained high numbers of TMEM119$^+$HLA$^+$CD68$^+$ microglia and TMEM119$^-$HLA$^+$CD68$^+$ macrophages, most of them with enlarged and foamy morphology that is typical of the activated and phagocytic state (*Figure 4g,g' and g''* arrows and h, h' arrows).

*Mixed active-inactive lesion (slowly expanding lesion).* The edge of these lesions was characterized by a rim of dense TMEM119$^+$HLA$^+$ microglia (*Figure 4i,i'* arrow head and j, j' arrow head) and TMEM119$^-$HLA$^+$ macrophages (*Figure 4i,i'* arrows and j, j', j'' arrows), both with obvious enlarged CD68$^+$ lysosomes (*Figure 4j''* arrows). Only a few HLA$^+$CD68$^+$ cells were present in the inactive lesion core and microglia showed profound reduction in the HLA signal (*Figure 4i,i'* arrows and j, j', j'' arrows).

## Qualitative assessment of t cells in staged ntz-rebound lesions by imc

Next, we analysed key molecules that differentiate between the phenotype and functional status of T cells in relation to the lesional stage and demyelinating activity of NTZ rebound lesions.

### Control subject white matter

In the white matter from a control subject CD3$^+$CD8$\alpha^-$ T cells or CD3$^+$CD8$\alpha^+$ T cells were rare of absent (*Figure 5a,a' and b,b'*).

*Normal-appearing white matter.* In the normal-appearing white matter, we identified some CD3$^+$-CD8$\alpha^-$ T cells (*Figure 5c,c'* arrow) and some CD3$^+$CD8$\alpha^+$ T cells (*Figure 5c,c'* dotted arrow) that did not show signs of activation as defined by the expression of NFAT1 which translocates to the nucleus of T cells upon T cell receptor activation (*Ma et al., 2015*) (*Figure 5d,d'* arrow and dotted arrow).

(P)reactive lesions. Within the (p)reactive lesions, CD3$^+$CD8$\alpha^-$ and CD3$^+$CD8$\alpha^+$ T cells were both prominent (*Figure 5e,e'* arrow and dotted arrow, respectively) but did not stain for NFAT1 and were therefore presumably not activated (*Figure 5f,f'* arrow and dotted arrow, respectively).

Active lesions. Active lesions contained both, CD3$^+$CD8$\alpha^-$ and CD3$^+$CD8$\alpha^+$ T cells (*Figure 5g,g'* arrow and dotted arrow, respectively), mostly located in the perivascular area (*Figure 5g'* inset), but also scattered in the parenchyma. Some CD3$^+$CD8$\alpha^+$ T cells were also activated based on the expression of NFAT1 (*Figure 5h,h'* arrow head).

Mixed active-inactive lesion (slowly expanding lesion). Similar to the core of active lesions, the edge of the slowly expanding lesions contained both, CD3$^+$CD8$\alpha^-$ and CD3$^+$CD8$\alpha^+$ T cells (*Figure 5i,i'* arrow and dotted arrow, respectively). A few CD3$^+$CD8$\alpha^-$ T cells and some CD3$^+$CD8$\alpha^+$ T cells were also NFAT1$^+$ (*Figure 5j,j'* white arrow and yellow arrow head, respectively). These were found both in the perivascular area and in the parenchyma (*Figure 5j''*). The staining pattern of NFAT1 was consistent with the nuclear localization of this transcription factor (*Figure 5k–n*). Nuclear NFAT1 signal was also observed on CD3$^-$ cells, consistent with reports of its localization of cells other than T cells (*Ma et al., 2015*) (*Figure 5h,h' and j,j'* white arrows head). Occasionally, we observed Ki67$^+$ proliferating cells (*Figure 5o*, arrow and inset), some of which were CD3$^+$ T cells (*Figure 5o*, dotted arrow). In the inactive lesion core, we observed both scattered CD3$^+$CD8$\alpha^-$ T cells and CD3$^+$CD8$\alpha^+$ T cells (*Figure 5i*).

## Qualitative assessment of B cells in staged NTZ-rebound lesions by IMC

Next we analysed key molecules that differentiate between the phenotype of B cells in relation to the lesional stage and demyelinating activity of NTZ rebound lesions.

### Control subject white matter

IgM staining on cells can be indicative of either naive B lymphocytes or IgM memory B cells. Therefore we first analysed the tissue for the presence of B cell-associated IgM. In control white matter, IgM was not found in association with Ig$\kappa$/Ig$\lambda^+$ B cells in the parenchyma but was only found in association with blood vessels which were identified by collagen staining. Further analysis showed that the IgM signal in the perivascular space co-localizes with immunoglobulin light chain Ig$\kappa$/Ig$\lambda$, indicating that this IgM$^+$ Ig$\kappa$/Ig$\lambda^+$ signal represents either naive or IgM memory B cells, or alternatively cell-free immunoglobulins (*Figure 6a,a' and b,b'* arrow head).

### Normal-appearing white matter

In the normal-appearing white matter, the IgM signal was found both in association with blood vessels (*Figure 6a,a' and b,b'* arrow head) and with nucleated Ig$\kappa$/Ig$\lambda^+$ B cells scattered in the parenchyma (*Figure 6a,a' and b,b'* dotted arrows). Nucleated Ig$\kappa$/Ig$\lambda^+$ B cells that were IgM$^-$ were also found in the parenchyma (*Figure 6a,a' and b,b'* solid arrows) indicating the presence of class switched B cells.

(P)reactive lesions. Within the (p)reactive lesions, IgM was exclusively detected within blood vessels (*Figure 6e,e'* arrow head). Nucleated Ig$\kappa$/Ig$\lambda^+$ B cells were present and expressed a switched B cell phenotype (IgM$^-$) (*Figure 6e,e' and f,f'* solid arrows).

Active lesions. Within the active lesions, nucleated Ig$\kappa$/Ig$\lambda^+$ B cells that displayed a switched phenotype (IgM$^-$) were mostly present (*Figure 6g,g' and h,h'* solid arrows).

Mixed active-inactive lesion (slowly expanding lesion). Similarly to active lesions, at the edge of mixed active-inactive lesions, nucleated Ig$\kappa$/Ig$\lambda^+$ B cells were present and displayed a switched memory phenotype (IgM$^-$) (*Figure 6i,i' and j,j'* solid arrows).

## 15-plex quantification of immune cells in staged NTZ rebound lesions by IMC

Following the visualization of markers of interest in tissue sections by IMC, we pre-set thresholds for each marker and analysed combinations of markers that identify the phenotype and functional status of immune cells, as shown in *Figure 7—figure supplement 5*. Focusing on the NTZ rebound tissue, we assembled these data into heat maps to visualize quantitatively the cellular content of each region of interest (*Figure 7a*).

*Macrophages and microglia.* CD45$^{high}$HLA$^+$TMEM$^-$ macrophages were found in the normal-appearing white matter and a low density of macrophages (3 cells/mm$^2$) contained PLP within their CD68$^+$ lysosomes, indicative of demyelinating activity. The number of demyelinating CD45$^{high}$HLA$^+$-TMEM$^-$CD68$^+$PLP$^+$ macrophages drastically increased in the (p)reactive lesions (34.8 cells/mm$^2$), reaching peak density in the core of active lesions (36.5 cells/mm$^2$) and edge of active-inactive demyelinating lesions (50.5 cells/mm$^2$). In the active core we also found a high density (92.6 cells/mm$^2$) of CD45$^{high}$HLA$^+$TMEM$^-$CD68$^+$PLP$^-$ macrophages, which represent phagocytes with enlarged but empty vacuoles.

Similarly to demyelinating CD45$^{high}$HLA$^+$TMEM$^-$CD68$^+$PLP$^+$ macrophages, demyelinating CD45$^{low}$HLA$^+$TMEM$^+$CD68$^+$PLP$^+$ microglia were found in high numbers in the (p)reactive lesions with peak density in the core of active lesions (249.5 cells/mm$^2$) and edge of active-inactive demyelinating lesions (403.3 cells/mm$^2$). Also in line with the distribution of non-demyelinating macrophages, a high density (224.9 cells/mm$^2$) of non-demyelianting CD45$^{low}$HLA$^+$TMEM$^+$CD68$^+$PLP$^-$ microglia with empty vacuoles were found in the core of active lesions. Overall, we found that in the core of active lesions on average 79% of HLA$^+$ cells are microglia and that they constitute 87% of the actively demyelinating (PLP$^+$) phagocytes. In the edge of a mixed active-inactive demyelinating lesion, we found that on average 88% of HLA$^+$ cells are microglia and that they constitute 89% of actively demyelinating (PLP$^+$) phagocytes.

*T cells.* Both CD45$^{high}$CD3$^+$CD8α$^+$CD4$^-$ (CD8) T cells and CD45$^{high}$CD3$^+$CD8α$^-$CD4$^+$ (CD4) T cells were abundant in the NTZ rebound tissue from the (p)reactive lesional stage (CD8$^+$ T cells, 53.4 cells/mm$^2$, CD4$^+$ T cells, 49.7 cells/mm$^2$) with peak densities in the core of active lesions (CD8$^+$ T cells, 58.6 cells/mm$^2$, CD4$^+$ T cells, 38.5 cells/mm$^2$), the periplaque (CD8$^+$ T cells, 92.3 cells/mm$^2$, CD4$^+$ T cells, 32.2 cells/mm$^2$) and the rim (CD8$^+$ T cells, 43.2 cells/mm$^2$, CD4$^+$ T cells, 68.8 cells/mm$^2$) of mixed active-inactive lesions. Overall, we found that in the core of active lesions on average 60% of T cells are CD8$^+$, 3% of which are activated and proliferating (NFAT$^+$Ki67$^+$). On the contrary, in the edge of a mixed active-inactive lesion, we found that on average 61% of T cells are CD4$^+$, 17% of which are activated and proliferating (NFAT$^+$Ki67$^+$). We verified these findings by examining an independent combination of markers – co-expression of CD38 and HLA on both CD4$^+$ and CD8$^+$ T cells is associated with T cell activation in the context of viral infection (*Wang et al., 2018*). We found that CD4$^+$CD38$^+$HLA$^+$ and CD8$^+$CD38$^+$HLA$^+$ were likewise enriched in the core of the active lesion and the edge of the active/inactive lesion with CD4$^+$CD38$^+$HLA$^+$ T cells being particularly represented at the edge of the active/inactive lesion. However unlike the NFAT$^+$Ki67$^+$ T cells, CD38$^+$-HLA$^+$ T cells were present in particularly high density in the (p)reactive lesion.

In addition to conventional CD4$^+$ and CD8$^+$ T cells, in this NTZ rebound patient we also identified a subpopulation of T cells that we defined 'unclassified' since they didn't fall into the classical CD45$^{high}$CD3$^+$CD8a$^+$CD4$^-$ T cell and CD45$^{high}$CD3$^+$CD8a$^-$CD4$^+$ T cell populations but were either double positive (CD45$^{high}$CD3$^+$CD8a$^+$CD4$^+$) or double negative (CD45$^{high}$CD3$^+$CD8a$^-$CD4$^-$) for CD8a and CD4 (see *Figure 7—figure supplement 6*). We found that unclassified T cells were present from the (p)reactive lesional stage (11.2 cells/mm$^2$, 10% of all detected T cells) with peak densities in the core of active lesions (25.1 cells/mm$^2$, 21% of all detected T cells) and at the rim (50.1 cells/mm$^2$, 31% of all detected T cells) of mixed active-inactive lesions.

*B cells.* Using the CD38 marker, we were able to further define B cells sub-populations beyond the qualitative images in *Figure 6*. We found B cells across all lesion types with switched memory CD45$^{high}$Igκ/Ig λ$^+$IgM$^-$CD38$^+$ B cells predominating in the core of active lesions (29.2 cells/mm$^2$, 58% of all detected B cells) and periplaque white matter (22.2 cells/mm$^2$, 87% of all detected B cells) and at the lesion rim (44.0 cells/mm$^2$, 44% of all detected B cells) of mixed active inactive lesions.

## Analysis of the distribution of immune cells in staged NTZ-rebound lesions by IMC

Since the distribution of blood-derived immune cells in relation to blood vessels can inform on the relationship between immune infiltrates and tissue injury, we performed a morphometric analysis of the distance between functional cell types and blood vessels in different NTZ rebound lesion areas (*Figure 7b*).

*Macrophages.* We found that demyelinating CD45$^{high}$HLA$^+$TMEM$^-$CD68$^+$PLP$^+$ macrophages infiltrated the lesion parenchyma in (p)reactive lesions (average distance from blood vessels, 136 μm) and periplaque white matter (average distance from blood vessels, 131–191 μm), indicating that

demyelinating events occur already in tissue that does not show obvious signs of demyelination. Demyelinating macrophages were mostly found in close proximity to blood vessels in active demyelinating lesions (average distance from blood vessels, 87 µm) and at the edge of active-inactive demyelinating lesions (average distance from blood vessels, 70 µm). Non-demyelinating CD45$^{high}$HLA$^+$TMEM$^-$CD68$^+$PLP$^-$ macrophages were found within the lesion parenchyma in both active lesions (average distance from blood vessels, 107 µm) and active-inactive lesions (edge: average distance from blood vessels, 244 µm; core:average distance from blood vessels, 279 µm), representing phagocytes that are no longer actively demyelinating.

*T cells.* In (p)reactive and periplaque white matter, both CD8$^+$ and CD4$^+$ T cells infiltrated the parenchyma (CD8$^+$ T cells: average distance from blood vessels, 114–238 µm; CD4$^+$ T cells: average distance from blood vessels, 80–208 µm). At the edge (CD8$^+$ T cells: average distance from blood vessels, 45 µm; CD4$^+$ T cells: average distance from blood vessels, 21 µm) and core (CD8$^+$ T cells: average distance from blood vessels, 121 µm; CD4$^+$ T cells: average distance from blood vessels, 89 µm) of active-inactive lesions, CD4$^+$ T cells were located in closer proximity to blood vessels compared to CD8$^+$ cells, which instead appeared to diffusely infiltrate the lesional parenchyma. CD8$^+$ and CD4$^+$ T cells were found to equally infiltrate the parenchyma in active lesions (CD8$^+$ T cells: average distance from blood vessels, 89 µm; CD4$^+$ T cells: average distance from blood vessels, 88 µm).

*B cells.* We found that naïve CD45$^{high}$Igκ/Igλ$^+$IgM$^+$CD38$^-$ B cells and switched memory CD45$^{high}$Igκ/Igλ$^+$IgM$^-$CD38$^+$ B cells infiltrated the parenchyma in both (p)reactive lesions (naïve B cells: average distance from blood vessels, 62 µm; memory switched B cells: average distance from blood vessels, 92 µm) and periplaque white matter (naïve B cells: average distance from blood vessels, 71 µm; memory switched B cells: average distance from blood vessels, 103–113 µm). Within lesions, naïve B cells were focally located in the perivascular space of veins at the the edge (average distance from blood vessels, 2 µm) and core (average distance from blood vessels, 9 µm) of active-inactive lesions. Switched memory CD45$^{high}$Igκ/Igλ$^+$IgM$^-$CD38$^+$ B cells were present in the vicinity of blood vessels at the rim (average distance from blood vessels, 44 µm) and core (average distance from blood vessels, 19 µm) of active-inactive lesions but were found to also diffusely infiltrate the parenchyma of active lesions (average distance from blood vessels, 82 µm).

## Analysis of Potential of Heat-diffusion Affinity-based Transition Embedding (PHATE) mapping

To study the dynamics of cell phenotypes, Potential of Heat-diffusion Affinity-based Transition Embedding (PHATE) mapping was performed. We focused specifically on T cells because previous studies have examined the localization of T cells with respect to blood vessels (*Machado-Santos et al., 2018*), and we were likewise interested in the large population of 'unclassified' T cells uncovered using the IMC approach. Using PHATE, we found that CD4$^+$ and CD8$^+$ T cells segregated in their own cluster. However, interestingly there is a population CD4$^+$, CD8$^+$ and unassigned T cells that form a distinct cluster in this particular NTZ rebound patient. Unassigned T cells are enriched within this distinct cluster (*Figure 8a*). We call this cluster 't1'. Further, we noted that mixed active-inactive lesions are the most likely lesion type to contain Cluster t1 cells in this particular NTZ rebound patient (*Figure 8b*). Based on distance from Collagen$^+$ blood vessels, Cluster t1 are almost exclusively found close to blood vessels whereas the CD4$^+$ and CD8$^+$ T cells that discriminate themselves from Cluster t1 cells are farther from the blood vessels. In addition, we observed that Cluster t1 cells are more likely to have higher levels of CD45 and are less likely to have low levels of HLA (a marker of T cell exhaustion) but seem to not be distinguishable from other lesion-resident cells in terms of Ki67, NFAT or CD38 expression (*Figure 8c*). Thus, using PHATE, we were able to discern a distinct cluster of T cells that were enriched within active-inactive lesions in this particular NTZ rebound patient.

## Discussion

In this study we simultaneously applied 15+ metal-conjugated antibodies to brain tissue specimens and used IMC and analytical pipelines to ascertain the phenotype and location of multiple immune cell subsets. The antibody panel contained both cell-specific and functional markers, and enabled the analysis of single-cell phenotypes and functional states of resident microglia, blood-derived

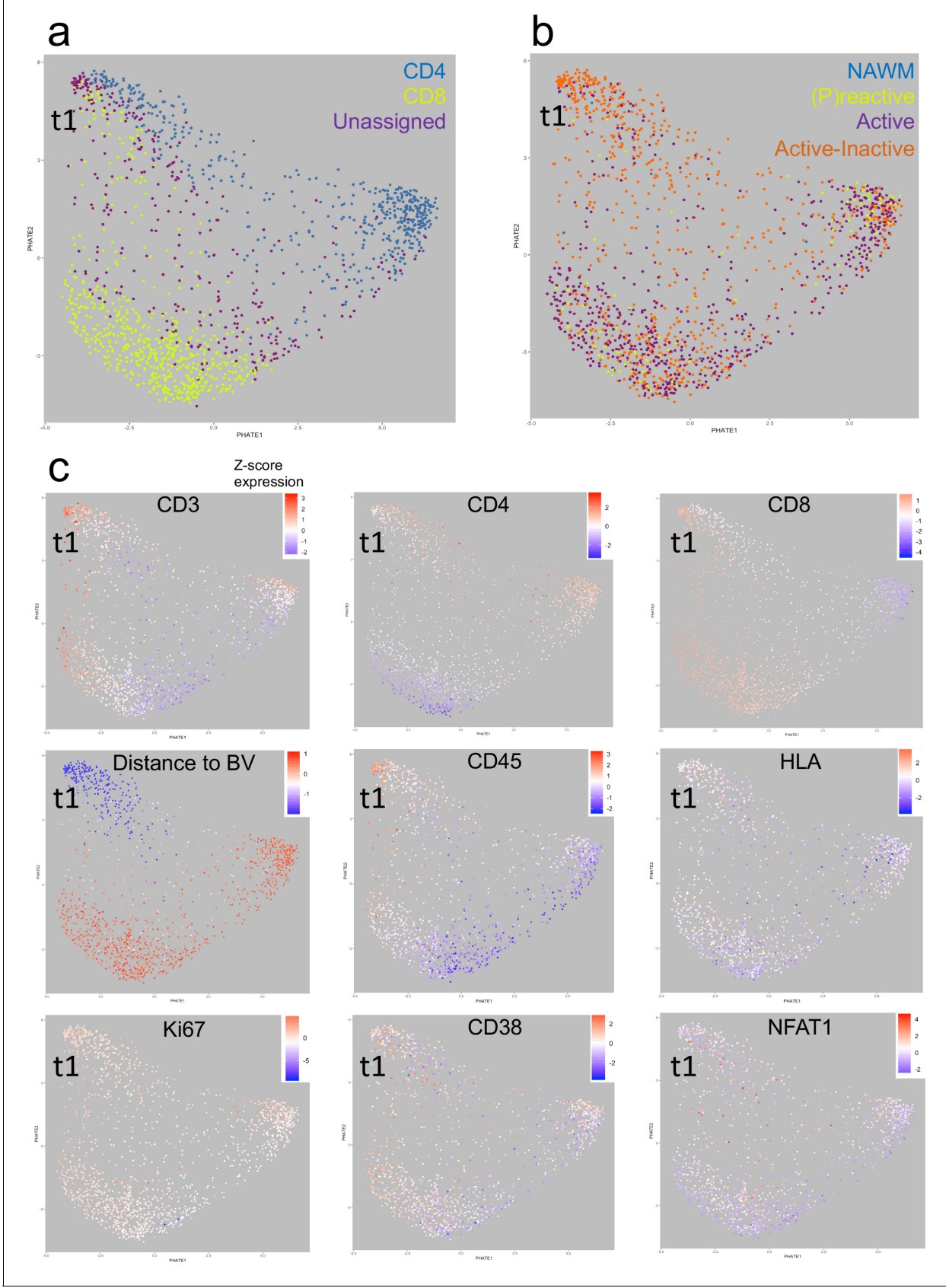

**Figure 8.** Heat-diffusion Affinity-based Transition Embedding (PHATE) mapping of T cells. (a–c) PHATE plots of all T cells analyzed in this study, colored by (a) cell class, (b) lesion type of residence and (c) relative marker expression intensity or distance to blood vessels. The heatmap scales in (c) represent the range of Z-score normalized values for a given parameter. NAWM = normal appearing white matter; BV = blood vessels; t1 = identified T cell cluster (see text for explanation).

DOI: https://doi.org/10.7554/eLife.48051.022

(recruited) macrophages, T and B lymphocytes in demyelinating and highly inflammatory lesions in a case of severe rebound MS disease activity after NTZ cessation (*Larochelle et al., 2017*). We first showed the validity of the technology on post-mortem brain tissue from the case of NTZ rebound disease activity and then applied it to the analysis of immune cells in lesions compared to control brain tissue.

IMC reproduced IHC- and IF-equivalent staining patterns with no apparent changes in specificity compared to standard IF. Therefore, antibodies validated with IF for the study of the MS brain will likely be applicable to the IMC approach. It should be noted, however, that the concentration and the staining conditions of some IHC- or IF-verified antibodies may not be implemented as is into the IMC protocol: titration and/or amplification (for example with biotin-streptavin) of pathologist-verified antibodies is required for optimal visualization by IMC.

In addition to visualizing a multitude of cell types, IMC allows for inclusion and exclusion criteria of selected markers to provide better confidence of cell identity. Furthermore, the highly quantitative nature of the IMC approach enables the analyses of data with pre-set thresholds for each marker, and permits further validation based on combinations of other markers. For example, we were able to distinguish CD45[high] cells that were TMEM119[-]CD68[+] thus identifying macrophages *versus* CD45[low/+] cells that were TMEM119[+]CD68[+] thus identifying microglia.

We found that in this particular control brain, microglial cells lose their homeostatic phenotype and acquire an activated state. This is in line with an earlier study demonstrating expression of certain activation markers by microglia within the normal human brain, and it is in agreement with recent immunohistological findings that show no expression of the homeostatic molecule P2RY12 (*Vogel et al., 2013*) in 48% of microglia in control brains (*Zrzavy et al., 2017*). Whether this activation state is the result of systemic exposure to recurrent infections (*Perry et al., 2010*) or is the result of vascular and neurodegenerative changes related to normal aging (*Conde and Streit, 2006*) (the control subject was 86 years old), or whether it is an inherent property of microglia in the human brain (*Zrzavy et al., 2017*), is unclear.

In line with recent observations in carefully staged lesions from a large cohort of MS patients at well-defined disease stages (*Zrzavy et al., 2017*), we found that in this particular NTZ-rebound case microglial activation was not restricted to lesional tissue but was also present in the normal-appearing white matter and (p)reactive lesion site. In these regions, although myelination appeared normal, we also found that demyelinating blood-derived macrophages infiltrated the parenchyma. In active lesions and in the active edge of mixed active-inactive demyelinating lesions (*Kuhlmann et al., 2017*) (slowly expanding or 'smouldering'; see *Frischer et al., 2009*), microglia and macrophages displayed similar phenotypic changes characterized by the predominant expression of markers associated with activation and phagocytosis. In contrast, in the core of mixed active-inactive demyelinating lesions, microglia and macrophages lost expression of molecules involved in antigen presentation and drastically reduced their phagocytic activity, as previously described (*Zrzavy et al., 2017*). Notably, a large proportion (on average 88%) of demyelinating myeloid cells in active lesions and at the edge of mixed active-inactive lesions were derived from the resident microglial pool, whereas macrophages that infiltrated the parenchyma of these lesional areas were largely inactive as indicated by the presence of enlarged but empty vacuoles in these cells. This is likely the result of the macrophage's inability to digest the myelin's neutral lipid components that accumulate and persist in macrophages.

In terms of lymphocytes, in classical active lesions and mixed active-inactive demyelinating lesions from this particular NTZ-rebound case T cells were abundant. Although CD8[+] T cells generally predominated across lesional stages, and in some cases proliferated (as also shown in a recent study by *Machado-Santos et al., 2018*), we also found a conspicuous number of CD4[+] T cells not only within lesions but also at the (p)reactive lesion site and periplaque white matter, likely reflecting the highly inflammatory nature of the lesions in this NTZ-rebound case. In addition,

CD4$^+$CD38$^+$HLA$^+$'chronically activated' T cells (*Wang et al., 2018*) were also particularly abundant in the (p)reactive lesion site. This suggests an involvement of these cells in the early stages of lesion formation, even in established lesions in this particular case. Similar to other findings in the case of T lymphocytes, our data also reproduced immunohistological findings that described B lymphocytes in all lesion stages in lower numbers compared to T cells (*Machado-Santos et al., 2018*). By using IgM in combination with CD38 and κ/λ, our panel has the increased capacity of identifying different B cell subsets, including IgM$^+$ and switched memory B cells.

It has been suggested that CD8$^+$ T cells in lesions from patients with relapsing, progressive and fulminant acute MS show features of tissue-resident memory cells and play a central role in the establishment of tissue-specific immunological memory, propagating chronic compartmentalized inflammation and tissue damage in the MS brain by local activation following re-exposure to their cognate antigen (*Machado-Santos et al., 2018*). B cells are also detected in all MS lesion types but their localization seems to be restricted to the pervascular space of some veins (*Machado-Santos et al., 2018*). Machado-Santos et al have shown that the majority of B and T cells are present in the perivascular cuffs, distant from sites of initial myelin damage (*Machado-Santos et al., 2018*). This supports the possibility that demyelination is induced by soluble factors produced by lymphocyte which diffuse into the tissue and in turn activate phagocytes. Our findings from a single NTZ-rebound case with high inflammatory activity observed both perivascular localization of B and T cells, as well as examples of B and T cells that diffusely infiltrate the lesion parenchyma. However, due to the nature of the acquired region of interest, which doesn't capture the areas surrounding the site of ablation, it is possible that blood vessels were positioned immediately outside the region of interest. These would be missed in the cell-blood vessel distance analysis. Further analysis of larger areas across multiple tissue samples is required to definitely determine whether B and T cells are found within the lesion parenchyma as a generalized feature of MS.

A comprehensive phenotypic characterization of B cells in MS tissue is lacking and the role of B cells in MS lesions is currently unresolved. Recent clinical studies have reported a protective effect of therapies targeting CD20$^+$ B cells in MS patients, suggesting a major role for B cells in the disease process (*Hauser et al., 2017*; *Hauser et al., 2008*). Our IMC results allowed for better segregation of B cell phenotypes (memory, class switched etc). Further addition of other markers, particularly for plasma cells (such as CD138, TACI) will be important for a full characterization of B cell subsets within the MS brain. This is particularly relevant in light of recent findings that demonstrate that some B lineage cells play a protective role in neuroinflammatory processes (*Rojas et al., 2019*).

While IMC has the advantage of multiplexing capability, it also has limitations. For example, it yields information only about the brain region imaged and is low throughput. It is therefore possible that different cell populations can exist in brain regions and sublesional areas other than those imaged. As is the case for the analysis of tissue stained with standard IHC/IF methods, multiple regions must be acquired. In addition, our study has the limitation that it is based on the analysis of immune cells in lesions from a single case with severe MS disease activity after NTZ withdrawal. Our goal was to provide proof that the IMC technology can be used as a powerful tool for the analysis of complex cellular phenotypes in heterogeneous tissues such as the MS brain, and a future direction will be to use the technique on a well-characterized MS brain tissue cohort. Moreover, given that the cells in these lesions had known phenotypes, the supervised approach for thresholding used herein was reasonable. However in the future, unsupervised analysis of data sets generated using the IMC or other multiparametric in situ analysis approaches (*Li et al., 2017*; *Gerner et al., 2012*; *Moreau et al., 2012*; *Tsujikawa et al., 2017*; *Carvajal-Hausdorf et al., 2019*; *Wang et al., 2019*), may identify novel cell types in tissue that are understudied, for example the MS meninges. In addition, discovery of novel cell types using a technique such as IMC can then be recapitulated with standard techniques using multiple well-characterized specimens from established brain banks.

Overall, our data reproduced immunohistological patterns of microglia and lymphocyte activation that had previously been described in carefully staged MS brain lesions at well-defined disease stages (*Machado-Santos et al., 2018*; *Zrzavy et al., 2017*) using a multi-parameter approach. We propose that IMC will enable a high dimensional analysis of single-cell phenotypes along with their functional states, as well as cell-cell interactions in relation to lesion morphometry and (demyelinating) activity. The IMC approach in combination with parallel high throughput techniques has the potential to profoundly impact our knowledge of the nature of the inflammatory response and tissue injury in the MS brain.

# Materials and methods

**Key resources table**

| Reagent type (species) or resource | Designation | Source or reference | Identifiers | Additional information |
|---|---|---|---|---|
| Antibody | Anti-Nucleic Acid-Ir191/Ir193 | Fluidigm | Cat#:201192A RRID: AB_2810850 | IMC: (1/3000) |
| Antibody | Anti-Proteolipid Protein-141Pr (Mouse monoclonal) | Bio-Rad | Cat#: MCA839G RRID:AB_2237198 | IMC: (1/25), IF: (1/25), IHC: (1/100) |
| Antibody | Anti-human CD38-167Er (Mouse monoclonal) | Fluidigm | Cat#:3167001B RRID: AB_2802110 | IMC: (1/2000) |
| Antibody | Anti-human CD45-154Sm (Mouse monoclonal) | Fluidigm | Cat#: 3154001B RRID:AB_2810854 | IMC: (1/2000) |
| Antibody | Anti-human CD68-159Tb (Mouse monoclonal) | Fluidigm | Cat#: 3159035D RRID:AB_2810859 | IMC: (1/100) |
| Antibody | Anti-human HLA-147Sm (Mouse monoclonal) | Fluidigm | Cat#: Ab55152 RRID: AB_944199 | IMC: (1/100), IF: (1/50), IHC: (1/100) |
| Antibody | Anti-human TMEM119-155Gd (Rabbit polyclonal) | Sigma-Aldrich | Cat#: HPA051870 RRID: AB_2681645 | IMC: (1/50), IF: (1/100) |
| Antibody | Anti-human CD3-170Er (Mouse monoclonal | Fluidigm | Cat#: 3170001 RRID: AB_2661807 | IMC: (1/100) |
| Antibody | Anti-human CD4-176Yb (Mouse monoclonal) | BioLegend | Cat#:344602 RRID: AB_1937277 | IMC: (1/100), IF: (1/20) |
| Antibody | Anti-human CD8a-162Dy (Mouse monoclonal) | Fluidigm | Cat#: 3162015B RRID:AB_2661802 | IMC: (1/100) |
| Antibody | Anti-human Granzyme B-171Yb (Mouse monoclonal) | ThermoFisher Scientific | Cat#: MA1-80734 RRID:AB_931084 | IMC: (1/25), IF: (1/20) |
| Antibody | Anti-human IgKappa-160Gd (Mouse monoclonal) | Fluidigm | Cat#:3160005B RRID:AB_2810855 | IMC: (1/3000) |
| Antibody | Anti-human IgLambda-151-Eu (Mouse monoclonal) | Fluidigm | Cat#: 3151004B RRID:AB_2810853 | IMC: (1/3000) |
| Antibody | Anti-human IgM-172Yb (Mouse monoclonal) | Fluidigm | Cat#: 3172004B RRID:AB_2810858 | IMC: (1/500) |
| Antibody | Anti-human Collagen Type I-169Tm (Goat polyclonal) | Fluidigm | Cat#: 3169023D RRID:AB_2810857 | IMC: (1/4000) |
| Antibody | Anti-human CD31-145Nd (Mouse polyclonal) | LSBio | Cat#: LS-C390863 RRID:AB_2810860 | IMC: (1/100) |
| Antibody | Anti-human NFAT1-143Nd (Rabbit monoclonal) | Fluidigm | Cat#: 3143023A RRID:AB_2810851 | IMC: (1/50) |
| Antibody | Anti-human Ki67-168Er (Mouse monoclonal) | Fluidigm | Cat#: 3168001B RRID:AB_2810856 | IMC: (1/100) |

## Patient case report and pathologic analysis of the brain

The clinical and pathologic characteristics of the case reported in this study have been previously published (*Larochelle et al., 2017*). Briefly, the patient was a 32-year-old female, diagnosed with relapsing-remitting MS in 2005. Natalizumab (NTZ) therapy was initiated (expanded disability status

scale (EDSS) 5.0 per relapse, and 3.5 upon induction on NTZ) but stopped after 2 years because, although clinically and radiologically stable (EDSS 2.0), the patient tested positive to the John Cunningham virus (JVC) virus antibody titer. Glatiramer acetate was started 1 month prior to NTZ cessation, and the patient received a 5 day course of intravenous (iv) methylprednisolone after the last NTZ infusion. Four months later, the patient was hospitalized for the presentation of new motor and cognitive deficits. Over the course of 2 weeks, the patient worsened (EDSS 9.5). Despite daily course of iv methylprednisolone, the patient developed several new gadolinium-enhancing lesions on repeated MRI. Since no clinical or radiological improvements were observed, the family decided to stop active care, as per patient's previous wishes. Autopsy was performed within 1 hr post-mortem, 4 days after withdrawal of all medication.

The patient had previously provided written consent for post-mortem donation of the CNS to research (ethics committee approval number BH.07.001). Pathologic analysis of the brain revealed abundant, active demyelinating, and highly inflammatory MS lesions with immunological pattern II (IgG- and complement-mediated) (*Larochelle et al., 2017*), according to *Lucchinetti et al. (2000)*. Despite the extent of the inflammation, progressive multifocal leukoencephalopathy or immune reconstitution inflammatory syndrome (IRIS) were excluded, from a pathologic point of view, although a later study proposed that Epstein-Barr virus-associated IRIS could have been the possible cause of the fulminant MS relapse in this case (*Serafini et al., 2018*). Of note, immunohistochemistry and qPCR for JCV was reported negative. Finally, the neuropathologist-confirmed diagnosis of severe MS rebound inflammatory demyelinating activity after NTZ withdrawal (*Larochelle et al., 2017*). The non-neurological control case was an 86-year-old female that died of cardiac arrest. This control case was obtained from the Netherlands Brain Bank (VU Medical Center ethic committee approval Reference number 2009/148).

## Sample characterization

Our study was performed on two frozen tissue blocks from the NTZ rebound case and one frozen tissue block from the non-neurological control case. We analysed immunopathologic changes in the white matter of the MS case, focusing on the following regions of interest (ROI) in lesions staged according to *Kuhlmann et al. (2017)* (see *Figure 1—figure supplement 1*): normal-appearing white matter (NAWM), located >1 cm distant from any lesions (detected in the block); the periplaque white matter (PPWM), located <1 cm distant from any lesions; (p)reactive lesions, that may represent the initial stage of a lesion (*Marik et al., 2007*; *Alvarez et al., 2015*; *Kuhlmann et al., 2017*), defined by the presence of microglia/macrophages in the absence of (obvious) demyelination, as described by *Luchetti et al. (2018)*; early active demyelinating lesions defined by presence of microglia/macrophages with early (MOG) and late (PLP) myelin degradation products throughout the lesion, as described (*Brück et al., 1995*), and previously shown for this NTZ rebound case (*Larochelle et al., 2017*), supporting the abundance of demyelinating activity in this type of lesions; the active edge of mixed active-inactive demyelinating lesions defined as slowly expanding lesions or smouldering lesions by Frischer et al, that are normally present in the progressive stage of MS (*Frischer et al., 2015*) but have also been described in cases with acute MS (*Zrzavy et al., 2017*); and lastly the inactive center of a mixed active-inactive demyelinating lesion. The normal white matter of control (WMC) was analysed as a reference background for the immunopathologic composition of the lesions in the NTZ rebound case. The lesion types and ROI analysed are indicated in *Table 1*.

## Selection of inflammatory markers

To define inflammatory cells as a whole, we used CD45, a general marker for microglia, macrophages and lymphocytes, with CD45$^{low/+}$ indicative of microglia and CD45$^{high}$ indicative of macrophages and lymphocytes. For microglia we additionally used TMEM119, that is expressed on yolk sac-derived (resident) microglia but not on recruited blood-derived macrophages (*Satoh et al., 2016*; *Bennett et al., 2016*). Other markers were used to detect phagocytosis (CD68) and capacity for antigen presentation (human leukocyte antigen, HLA). All T cells were detected with the cellular marker CD3. CD8$\alpha$ detected MHC class I restricted T cells while CD4 detected MHC class II restricted T cells. All B cells were identified by the expression of either the kappa ($\kappa$) or lambda ($\lambda$) allelic variants of the immunoglobulin light chain. CD38 was used to detect a multifunctional

molecule expressed by leucocytes in general and involved in the activation of T cells and B cells. IgM was used to identify naïve and non-class switch memory B cells, in addition to detecting free immunoglobulins. Furthermore, microglia and T cells express the transcription factor nuclear factor of activated T cells (NFAT1), that translocates to the nucleus upon activation (*Crabtree and Olson, 2002*). Therefore, we determined the localization of NFAT1 as an additional activation antigen of $CD45^{low/+}TMEM119^+$ microglia and $CD45^{high}CD3^+CD8\alpha^+$ or $CD45^{high}CD3^+CD8\alpha^-$ T cells. We also used the Ki67 marker of cell proliferation and PLP to identify myelin. Blood vessels were identified using markers of extracellular matrix (collagen) and endothelial cells (CD31). Each antibody clone was first titrated for immunofluorescence staining in control and NTZ rebound tissue, according to the dilutions shown in the Key Resouces Table, prior to methial-conjugation and IMC application.

## Histology, immunohistochemistry (IHC), and immunofluorescence (IF)

Ten-micron frozen tissue sections were mounted on Superfrost Plus glass slides (Knittel Glass) and stored at −80 °C until they were stained.

*Histology.* On the day of the staining, the slides were brought at room temperature and post-fixed in 10% formalin for 3 hr. Tissue sections were stained with Hematoxylin and Eosin (HE)/Luxol Fast Blue (LFB) to detect myelin lipids and Oil red O (ORO) to detect neutral lipids in phagocytic macrophages, as previously published (*Podjaski et al., 2015*).

*IHC.* On the day of the staining, the slides were brought to room temperature and post-fixed in ice-cold acetone for 10 min. Myelin protein was detected using an antibody for proteolipid protein (PLP) and microglia/macrophages were detected using an antibody for human leukocyte antigen (HLA). Endogenous peroxidases activity was blocked by incubation in PBS with 0.3% $H_2O_2$ for 20 min at room temperature. Non-specific protein binding was blocked by incubation with 10% normal goat serum (DAKO). Primary antibodies were applied overnight at 4°C, diluted in normal antibody diluent (Immunologic, Duiven, The Netherlands) according to dilutions noted in Key Resouces Table. The following day, sections were incubated with a post-antibody blocking solution for monoclonal antibodies (Immunologic) diluted 1:1 in PBS for 15 min at RT. Detection was performed by incubating tissue sections in secondary Poly-HRP (horseradish peroxidase)-goat anti-mouse/rabbit/rat IgG (Immunologic) antibodies diluted 1:1 in PBS for 30 min at RT followed by application of DAB (3,3-diaminobenzidine tetrahydrochloride (Vector Laboratories, Burlingame, CA, U.S.A.) as a chromogen. Counterstaining was performed with hematoxylin (Sigma-Aldrich) for 10 min. The sections were subsequently dehydrated through a series of ethyl alcohol solutions and then placed in xylene before being coverslipped with Entellan mounting media (Sigma Aldrich). The colorimetric staining was visualized under a light microscope (Axioscope, Zeiss), connected to a digital camera (AxioCam MRc, Zeiss) and imaged with Zen pro 2.0 imaging software (Zeiss).

*IF.IF.* On the day of the staining, the slides were brought to room temperature and incubated in ice-cold acetone for 10 min followed by 70% ethanol for 10 min to reduce the autofluorescence signal derived from the fatty myelin sheets. Slides were subsequently rehydrated in 0.05% PBS-tween

**Table 1.** Lesion types and regions of interest.

| Case | Tissue block (anatomical location) | Lesion type | Region of interest |
|---|---|---|---|
| C, 95–056 | A (superior frontal gyrus) | | WMC |
| MS, AB129 | CL3a (cerebellum) | | NAWM |
| | | 2x Active demyelinating (pattern II) | PPWM, center |
| | | 3x Mixed active/inactive demyelinating | PPWM, edge, core |
| | CR4a (cerebellum) | | NAWM |
| | | (p)reactive | |
| | | 3x Active demyelinating (pattern II) | center |
| | | 2x Mixed active/inactive demyelinating | edge, core |

C, control; MS, multiple sclerosis; WMC, white matter of control; NAWM, normal-appearing white matter; PPWM, periplaque white matter.

DOI: https://doi.org/10.7554/eLife.48051.023

for 10 min at room temperature followed by incubation in 10% normal goat serum (DAKO) to block nonspecific binding sites. Sections were then incubated overnight at 4 °C with primary antibody diluted in 3% normal goat serum (see dilutions in Key Resouces Table). Primary antibodies were detected using fluorochrome-conjugated secondary antibodies (Sigma-Aldrich) diluted 1:200 in 1% Triton-X100. Sections were incubated with DAPI (Sigma Aldrich) diluted 1:3000 to visualize the nuclei. Slides were washed in PBS, air dried and mounted in aqueous mounting medium. Using the appropriate filters, the IF signal was visualized with an Axio Imager Z1, Zeiss microscope connected to a digital camera (AxioCam 506 mono, Zeiss) and imaged with Zen pro 2.0 imaging software (Zeiss).

To control for antibody specificity, tissue sections were stained according to the IF or IHC protocols described above except for the primary antibody incubation step, which was omitted.

## Imaging mass cytometry

The work flow of imaging mass cytometry is shown in *Figure 1—figure supplement 2* and explained in detail below.

*Conjugation of antibodies with lanthanide metals.* Lanthanide metal-conjugated antibodies were either obtained from Fluidigm (Markham, Ontario, Canada) or conjugated at SickKids-UHN Flow and Mass Cytometry Facility (Toronto, Ontario, Canada), using the MaxPar X8 labelling kit from Fluidigm (catalogue number 201169B) as previously described (*Han et al., 2018*). Briefly, a purified carrier-free antibody was partially reduced with TCEP buffer (Fluidigm, catalogue number 77720) at 37°C. The reduced antibody was then incubated with an excess of metal-loaded MaxPar X8 polymer for 90 min at 37°C. The metal-labeled antibody was then recovered using a 50 kDa size exclusion filter. The percent yield of metal-conjugated antibody was determined by measuring the absorbance of the conjugate at 280 nm. The recovery of our metal-conjugated antibodies was 69–78%. Antibody stabilizer was then added to the metal-conjugated antibodies before long-term storage at 4°C.

*Staining for Imaging Mass Cytometry.* On the day of staining, the slides were brought to room temperature and rehydrated with 0.05% PBS-Tween in a humidified chamber for 20 min at room temperature. Non-specific protein binding was blocked by incubation with 10% normal goat serum for 20 min at room temperature followed by incubation with blocking solution (ThermoScientific Superblock Blocking Buffer in PBS) for 45 min at room temperature. A cocktail of primary antibodies, diluted in 0.5% BSA, was applied overnight at 4°C at the dilutions indicated in Key Resouces Table. The following day, slides were first washed with 0.05% PBS-Tween and then with PBS, followed by incubation with Iridium-conjugated intercalator (Fluidigm, catalogue number 201192B), diluted 1:2000 in 0.5% BSA for 30 min at room temperature. Lastly, slides were dipped in water (Invitrogen ultrapure distilled water), air dried and stored at room temperature until they were ablated.

*Identification of region of interest (ROI) for laser ablation.* Two serial sections each stained for either IF or IMC, were used. Based on IF staining with an antibody specific for proteolipid protein (PLP) (to visualize myelin) and DAPI (to visualize nuclei), ROIs were selected for ablation to capture the regions of interest for this study.

*High-spatial resolution laser ablation of tissue sections.* Tissue sections were analyzed by IMC, which couples laser ablation techniques and CyTOF mass spectrometry (*Bandura et al., 2009*) (Cytof software v6.7). Briefly, a UV laser beam ($\lambda$ = 219 nm) with a 1µmx1µm spot size is used to ablate the tissue. The laser rasters across the tissue at a rate of 200 Hz (200 pixels/s) with the requisitie energy to fully remove the tissue within the selected region of interest. The time needed to analyze each image of 1 mm$^2$, using the methodology described in this manuscript, is approximately 1 hr and 45 min. The ablated tissue is then carried by a stream of inert helium and argon gas into the Helios (a CyTOF system) where the material is completely ionized in the inductively coupled plasma. The ionized material then passes through high pass ion optics to remove ions with a mass less than 75amu before the ions enter the time of flight detector where they are separated based on their mass (*Bodenmiller et al., 2012*; *Bendall et al., 2011*).

*Data analysis and image visualization.* Images of each mass channel were reconstructed by plotting the laser shot signals in the same order they were recorded, line scan by line scan, generating pseudo-colored intensity maps of each mass channel. These data were examined using MCD Viewer (V.1.0.560, Fluidigm). For qualitative assessments, images remained at the automatic threshold generated by MCD Viewer, based on the on the 98th percentile of signal. For further analysis, data were exported from MCD Viewer as tiff files, and each channel was run through an individual analysis

pipeline in CellProfiler (*Carpenter et al., 2006*; *Jones et al., 2008*) (V3.185) in order to despeckle the image. Composite images were created for each ROI using Image J (V1.52a), and any changes to the brightness or contrast of a given marker were consistent across ROIs.

*Calculation of the limit of detection.* MCD Viewer was used to export text files acquired with the Hyperion IMC instrument (Fluidigm Inc, Markham ON), which were then converted to 32-bit single-channel TIFF images. The polygon tool within ImageJ 1.15 s was used to manually outline the ROI (white matter of control, normal-appearing white matter) or subROI (periplaque white matter, lesion edge, lesion core), manually identified on the bases of PLP, HLA and Iridium-intercalator signals. Gray matter was excluded from subsequent analysis. Each image was despeckled in Definiens Developer XD 2.7 (Definiens Inc, Munich, Germany), using a 2D gray-level morphological opening filter with kernel radius of 1. In addition, the intensities of each marker were normalized using a modified z score approach, in which the intensity of each pixel is divided by the sum of (mean intensity of the image plus 3 times the standard deviation of the pixels in the image) $I_{zs} = I/(\mu_{Im}+3*\sigma_{Im})$. This normalization approach has been previously used (*Ellingson et al., 2012*) and we found that it allows for a reliable comparison between IMC markers across different channels, with per-marker comparisons holding robustly across a 16-fold antibody dilution series (data not shown).

*Single-cell segmentation.* In order to define cells, we used a customized segmentation algorithm that took into account both the presence of nuclear DNA Iridium-intercalator as well as a set of markers of interest (see example in *Figure 7—figure supplement 1*). In brief, a Gaussian blur was applied to the DNA signal and the resulting blurred image was segmented to identify nuclear content (*Figure 7—figure supplement 1a*). Segmentation around the nuclei was expanded to simulate the cytoplasm, corresponding to individual cell areas, using a combination of threshold and watershed filters (*Figure 7—figure supplement 1b*). Next we interrogated the segmented image for the presence of specific markers or combinations of markers that are either biologically co-expressed, or whose expression is mutually exclusive, according to the combination of markers indicated in *Table 2*. If a nucleated cell was positive for a marker or a combination of markers (see example for CD3 in *Figure 7—figure supplement 1c*), the marker(s) signal was used to refine the initial nuclear segmentation. Nucleated cells that were not positive for any of the markers used, were segmented purely based on DNA signal and expanded to simulate the cytoplasmic area around the nucleus.

*Gating strategy for quantitative analysis of T cell, B cell, macrophage and microglial cell subsets.* Segmented cell exports of raw and normalized marker intensities for all channels in each region of interest were exported as a single csv file. The per-cell mean intensities of each marker combination, (see marker list in *Table 2*), were linearly rescaled for visualization purposes. 2D log-log biaxial scatterplots of these intensities were generated in Python (V3.6.8) using matplotlib (V3.0.3). A positive- and negative-gating strategy was applied to establish thresholds that identify particular cell types. Quadrants were set on pathologist-verified positive cells. In brief, ROI was examined in Definiens Developer XD 2.7. Cells were manually annotated by a pathologist, based on the expression of a biologically relevant set of markers to identify cells in each class of interest as defined below (see examples of manual selection to identify myeloid cells in *Figure 7—figure supplement 2*; to identify T cells in *Figure 7—figure supplement 3*; to identify B cells in *Figure 7—figure supplement 4*). These identified positive cells were superimposed to the 2D log-log scatterplots to definitively establish gates that would capture the appropriate positivity range of each cell phenotype as shown in *Figure 7—figure supplement 5*.

*For T cells*: All nucleated cells expressing Igκ, Igλ, IgM, CD68 and HLA were eliminated, as these markers are not expressed on T cells. Gates were established for CD3 and CD45 based on a 2D log-log scatterplot of these markers. Following the identification of T-lineage cells, the same procedure was performed for CD3 vs CD4 and CD3 vs CD8, resulting in the identification of two subpopulations: CD4$^+$ T cells and CD8$^+$ T cells. Thresholds for Ki67 (a marker of proliferation) and NFAT1 (a marker of activation) were established based on manually annotated CD3$^+$KI67$^+$ and CD3$^+$NFAT1$^+$ cells, as described above. All cell populations were validated by manual annotation as described above.

*For B Cells:* All nucleated cells expressing CD3, CD4, CD8 and CD68 were eliminated as these markers are not expressed on B cells. B cells were further identified by CD45 above the same threshold set for T cells. Scatterplot comparison for Igκ and Igλ intensity identified Igκ+ and Igλ$^+$ single-positive populations. Igκ$^+$Igλ$^+$ double positive cells were eliminated as artifactual, since the two allelic variants cannot co-exist on a given cell. Within Igκ+ or Igλ$^+$ B-lineage cells, we compared IgM to

**Table 2.** Antibody panels for the identification of functional cell types by IMC.

| Functional cell type | Antibody panel |
| --- | --- |
| Macrophages and microglia | |
| Macrophages | CD45$^{high}$HLA$^+$TMEM119$^-$ |
| Activated macrophages | CD45$^{high}$HLA$^+$TMEM119$^-$CD68$^+$PLP$^-$ |
| Demyelinating macrophages | CD45$^{high}$HLA$^+$TMEM119$^-$CD68$^+$PLP$^+$ |
| Microglia | CD45$^{low/+}$HLA$^+$TMEM119$^+$ |
| Activated microglia | CD45$^{low/+}$HLA$^+$TMEM119$^+$CD68$^+$PLP$^-$ |
| Demyelinating microglia | CD45$^{low/+}$HLA$^+$TMEM119$^+$CD68$^+$PLP$^+$ |
| T Cells | |
| CD8$^+$ T cells | CD45$^+$CD3$^+$CD8$\alpha^+$CD4$^-$ |
| Proliferating CD8$^+$ T cells | CD45$^+$CD8$\alpha^+$CD3$^+$CD4$^-$Ki67$^+$ |
| Activated CD8$^+$ T cells | CD45$^+$CD3$^+$CD8$\alpha^+$CD4$^-$NFAT$^+$ |
| Activated and proliferating CD8$^+$ T cells | CD45$^+$CD3$^+$CD8$\alpha^+$CD4$^-$NFAT$^+$Ki67$^+$ |
| 'Chronically activated' CD8$^+$ T cells | CD45$^+$CD3$^+$CD8$\alpha^-$CD8$^+$CD38$^+$HLA$^+$ |
| CD4$^+$ T cells | CD45$^+$CD3$^+$CD8$\alpha^-$CD4$^+$ |
| Proliferating CD4$^+$ T cells | CD45$^+$CD8$\alpha^-$CD3$^+$CD4$^+$Ki67$^+$ |
| Activated CD4$^+$ T cells | CD45$^+$CD3$^+$CD8$\alpha^-$CD4$^+$NFAT$^+$ |
| Activated and proliferating CD4$^+$ T cells | CD45$^+$CD3$^+$CD8$\alpha^-$CD4$^+$NFAT$^+$Ki67$^+$ |
| 'Chronically activated' CD4$^+$ T cells | CD45$^+$CD3$^+$CD8$\alpha^-$CD4$^+$CD38$^+$HLA$^+$ |
| B Cells | |
| Naïve B Cells | CD45$^+$Igκ/Igλ$^+$IgM$^+$CD38$^-$ |
| IgM memory B cells | CD45$^+$Igκ/Igλ$^+$IgM$^+$CD38$^+$ |
| Switched memory B cells | CD45$^+$Igκ/Igλ$^+$IgM$^-$CD38$^+$ |

DOI: https://doi.org/10.7554/eLife.48051.024

CD38 to determine the relative abundance of IgM$^+$ or CD38$^+$ cell subpopulations. All cell populations were validated by manual annotation as described above.

*For macrophages and microglia*: All nucleated cells expressing CD3, CD4, CD8, Igκ/λ and IgM were eliminated as these markers are not expressed on macrophages and microglia. Discrimination of the remaining cells was visualized in a scatterplot for TMEM119 and CD45. The threshold for TMEM119 positive signal was determined by comparison to TMEM119$^+$ microglial cells that were identified by manual observation, relative to other cell types. The threshold for CD45 high or low signal was determined by the comparison to manually identified TMEM$^-$ macrophages. Manually identified microglial cells were used to establish the lower limit of the CD45 quadrants. Cells that were low for both TMEM119 and CD45 were labeled 'other' and ignored from subsequent analysis. These latter cells, likely correspond to astrocytes, oligodendrocytes and other cell types. Both TMEM119$^+$-CD45$^{low/+}$ microglial cells and TMEM$^-$CD45$^{high}$ macrophages were further evaluated for HLA, CD68 and PLP (depicted as scatterplots), to differentiate microglia or macrophages that are either resting or activated/phagocytic/demyelinating. All cell populations were validated by manual annotation as described above.

*Generation of cell density map.* The gating strategy described above was confirmed by plotting the appropriately gated cell types for major lineage markers (see examples in *Figure 7—figure supplement 6*). Note that Igκ>Igλ in *Figure 7—figure supplement 6a* consistent with over-representation of κ$^+$ B cells in humans (*Koulieris et al., 2012*). Following this confirmation, the density of all relevant cell subtypes was computed within each biological region of interest. A heat map, generated using Seaborn (V0.9.0), displayed the cell counts per mm$^2$ of tissue.

*Generation of distance map.* To assess the location of identified cells relative to blood vessels, collagen$^+$ perivascular regions of >10 μm diameter and >800 μm$^2$ area were segmented, and the distance between cells of interest and the border of these perivascular regions was calculated.

Similarly to the cellular density calculations, average vessel distances corresponding to the mean of the per-cell vessel distance values were computed, expressed in µm, and presented as a distance heat map. Some regions did not contain any cells of a particular type, leading to undefined values for those particular regions and cell type combinations (presented as 'n/a', not applicable).

*Potential of Heat-diffusion Affinity-based Transition Embedding (PHATE) mapping.* To study the dynamics of T cell phenotypes, Potential of Heat-diffusion Affinity-based Transition Embedding (PHATE) mapping was performed in R (*Moon, 2019*). Relevant parameters were selected for analysis of T cells (CD3, CD4, CD8, CD38, CD45, HLA, Ki67, NFAT1, distance to blood vessels). Raw, mean single-cell marker intensity values extracted from segmented IMC images as well as measured distances to blood vessels were subjected to log10 transformation followed by Z-score normalization. These normalized single-cell parameter values along with metadata indexing the class and lesion type of residence for each cell served as input for PHATE mapping, as detailed in the online user guide. The PHATE algorithm was executed with k = 100 and other parameters left as their default specifications. Plots were generated using the R ggplot2 package (*Hadley, 2016*).

## Statistical analysis

Where statistical testing was possible, all tests were performed using Prism software (v5.01; Graph-PadSoftware, San Diego, CA). Data distribution was tested for normality. Because all variables were not normally distributed, possible correlations between the density of nuclei or the density of cells (number of cells / mm$^2$ of tissue) or the % stained area detected either by IF or by IMC, were investigated with the nonparametric Spearman rank correlation. Differences were considered significant at $p < 0.05$.

## Acknowledgements

This work was funded by a team grant from the MS Society Research Foundation to AP and JG, and the National Multiple Sclerosis Society Research Grant RR-1602-07777 to VR. We would like to thank the brain donors at Université de Montréal and the Netherlands Brain Bank.

## Additional information

### Competing interests

Valeria Ramaglia: received a consulting honorarium from EMD Serono. Olga Ornatsky, Eric C Swanson: is an employee of Fluidigm Inc. Jennifer L Gommerman: is a consultant for Roche (Canada) and currently holds grants with Novartis, EMD Serono and Roche. The other authors declare that no competing interests exist.

### Funding

| Funder | Grant reference number | Author |
| --- | --- | --- |
| National Multiple Sclerosis Society | RR-1602-07777 | Valeria Ramaglia |
| Multiple Sclerosis Society of Canada | | Jennifer L Gommerman Alexandre Prat |

The funders had no role in study design, data collection and interpretation, or the decision to submit the work for publication.

### Author contributions

Valeria Ramaglia, Conceptualization, Formal analysis, Supervision, Funding acquisition, Investigation, Visualization, Writing—original draft, Writing—review and editing; Salma Sheikh-Mohamed, Formal analysis, Investigation; Karen Legg, Validation; Calvin Park, David Pitt, Formal analysis, Performed the PHATE analysis; Olga L Rojas, Supervision; Stephanie Zandee, Eric C Swanson, Investigation; Fred Fu, Trevor D McKee, Formal analysis; Olga Ornatsky, Conceptualization, Resources; Alexandre

Prat, Conceptualization, Resources, Funding acquisition; Jennifer L Gommerman, Conceptualization, Supervision, Funding acquisition, Writing—review and editing

Author ORCIDs
Valeria Ramaglia https://orcid.org/0000-0002-9401-5988
Eric C Swanson https://orcid.org/0000-0001-8454-1207
David Pitt http://orcid.org/0000-0002-6407-9542
Alexandre Prat http://orcid.org/0000-0001-6188-0580
Jennifer L Gommerman https://orcid.org/0000-0003-4576-6168

Ethics
Human subjects: This work included the use of post-mortem brain tissue. Written consent for post-mortem donation of the CNS to research from the MS donor was obtained (ethics committee approval number BH.07.001). Ethical approval for the use of post-mortem brain tissue from the control donor of the Netherlands Brain Bank was obtained (VU Medical Center ethic committee approval Reference number 2009/148).

Decision letter and Author response
Decision letter https://doi.org/10.7554/eLife.48051.027
Author response https://doi.org/10.7554/eLife.48051.028

## Additional files

### Supplementary files
• Transparent reporting form
DOI: https://doi.org/10.7554/eLife.48051.025

### Data availability
All data generated and analysed during this study are included in the manuscript and supporting files. Source data file has been provided for Figure 7.

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
