## [Decision Letter]

Thank you for submitting your article "Multiplexed imaging of immune cells in staged multiple sclerosis lesions by mass cytometry" for consideration by *eLife*. Your article has been reviewed by three peer reviewers, one of whom is a member of our Board of Reviewing Editors, and the evaluation has been overseen by Satyajit Rath as the Senior Editor. The reviewers have opted to remain anonymous.

The reviewers have discussed the reviews with one another and the Reviewing Editor has drafted this decision to help you prepare a revised submission.

Summary:

In this manuscript, Ramaglia et al. elegantly utilize imaging mass cytometry (IMC), an innovative multiplex protocol to perform comprehensive and functional analysis of cell heterogeneity and their distribution within MS lesions in a patient. They were able to study 15 immune markers simultaneously. This work highlights how the combination of multiplexed tissue imaging and advanced computational tools allows for a comprehensive characterization of immune/glial cells within/around MS lesions, their functional state and cell-cell interactions in relation to lesion localization. Given the array of immune cells infiltrating the CNS during disease likes MS, this technological advance will allow for a more granular understanding of cell phenotypes within distinct CNS compartments and how distinct inflammatory states shape their function and interactions. It is important to note that this is a single case study involving a rebound case of MS disease activity, and the authors must be careful in their conclusions of any findings from this work.

The reviewers agree that although the number of samples analyzed is small, these studies are extremely promising because they describe a novel method to perform molecular histopathology. Recent studies have described this technology for multiplexing stainings and for phenotyping cells in the tissue context in other disease conditions, such as cancer (e.g. Giesen et al., 2014, Chang et al., 2017, Carvajal-Hausdorf et al., 2019, Wang et al., 2019, etc.) Overall, it is a solid piece of work from a technical point of view and describes the potential use of this technique. Thus the description and application of this potentially interesting technology is not new per se, but this technology may have implications for future studies aimed at characterizing and phenotyping inflammatory processes in MS brain sections with the best possible precision. We hope that you can address the following points in your revisions of the manuscript.

Essential revisions:

– While this is a proof of principle study, the case number used for this study is very low. Most if not all investigations were done on a single MS and one control case. Therefore, the discussion should be geared towards the unique neuropathology of this case and how the multiparametric findings provided by IMC contrast with those of conventional immunostainings. Apart of showing the feasibility on single cases of the technology, it remains unclear how generalizable and reproducible the described findings are. Of note, the 11 lesions studied come from a patient with severe rebound MS disease activity after natalizumab cessation. This pathology is characterized by heavy immune cell infiltration and activation that represents an extreme side of the inflammatory spectrum. Hence, the authors cannot claim that the majority of demyelinating macrophage-like cells in active lesions are derived from the resident microglial pool. Likewise, stating that B cells with a predominant switched memory phenotype are enriched in parenchyma across all lesion stages while unswitched B cells localized to the vasculature might only be true for anti-VLA-4 rebound disease, but not for MS. The authors also observed high numbers of CD4 and CD8 T cells within (p)reactive lesions, a surprising finding that could be confounded by the exacerbated neuroinflammation seen in this patient and that contrasts with the elevated T cell infiltration reported in active lesions. Thus, the authors would need a comprehensive characterization of MS lesions from prototypical MS specimens to raise such points conclusively. However, given the similarities in neuroinflammation between natalizumab cessation (including JCV seropositivity) and PML, the findings of this study might be more relevant for conditions like PML. This is further supported by the predominant perivascular and parenchymal infiltration of CD8^+^ T cells found in this patient.

– On the technical side, the manuscript needs to highlight the benefits of IMC compared to conventional immunostaining. The authors should consider including imaging data highlighting the benefits of IMC such as panels including more than 8 parameters at low and high power will help to appreciate the potential of this technique. Given that most analysis of phenotyping cells were finally made by 3-4 marker combinations it not sufficient elaborated what analyses are done here which are not possible with conventional fluoresecence co-immunostainings.

– The authors compared only a limited set of markers between IHC and IMC. How did the other markers perform between IHC vs. IMC? In particular, the sensitivity of IHC is strongly dependent on the amplification method used. Thus, stating that IMC might be more sensitive than IHC seems premature at this stage unless supported by a more systematic comparison.

– When performing the characterization of T cell populations, the authors described a large number of unclassified T cells. The total number of these cells should be included in Figure 7. In future studies, it would be of interest to include markers to determine which cell types are making up such large cohort. The study would benefit from making use of the novel computational tools available to analyze IMC data. Some examples include PHATE mapping and Pseudotime analysis which would allow to establish phenotypic fates within the immune cell populations studied.

– The data analysis workflow as described in this manuscript needs manual intervention at various steps which are not described in sufficient details. This may make it difficult to reproduce the procedure by others in the future.

– Despite the capacity to multiplex stainings for MS, it is not well elaborated what one can learn about the MS pathophysiology by this technology in the current study. The authors should elaborate here more, and this comment relates to the single case study above.

– In the Discussion the authors should consider the state of the literature regarding multiparametric in situ analysis (seminal papers include: Gerner et al., 2012, Moreau et al., 2012, Tsujikawa et al., 2017, Li et al., 2017 and Kwong et al., 2017) and explain how IMC compares to established techniques. As a whole, given the technical nature of the manuscript, the authors should base their conclusions on the experimental underpinnings of IMC and limit extrapolations to MS generally.

---

## [Author Response]

Essential revisions:– While this is a proof of principle study, the case number used for this study is very low. Most if not all investigations were done on a single MS and one control case. Therefore, the discussion should be geared towards the unique neuropathology of this case and how the multiparametric findings provided by IMC contrast with those of conventional immunostainings. Apart of showing the feasibility on single cases of the technology, it remains unclear how generalizable and reproducible the described findings are. Of note, the 11 lesions studied come from a patient with severe rebound MS disease activity after natalizumab cessation. This pathology is characterized by heavy immune cell infiltration and activation that represents an extreme side of the inflammatory spectrum. Hence, the authors cannot claim that the majority of demyelinating macrophage-like cells in active lesions are derived from the resident microglial pool. Likewise, stating that B cells with a predominant switched memory phenotype are enriched in parenchyma across all lesion stages while unswitched B cells localized to the vasculature might only be true for anti-VLA-4 rebound disease, but not for MS.

The reviewers are quite correct, and we apologize for over-stating our conclusions. We have changed the text as well as the Abstract, refraining from generalizing our observation using a single case of aggressive rebound to the spectrum of what can be found in more “classical” MS. Rather, we will explicitly refer to this case a Natalizumab (NTZ)-rebound case. We have provided context as to why this case was chosen (serves as a good “positive” control for neuroinflammation) in the last paragraph of the Introduction and in the first paragraph of the subsection “Comparability of IF versus IMC approach and specificity of metal-conjugated antibodies on brain-resident cell types”.

The authors also observed high numbers of CD4 and CD8 T cells within (p)reactive lesions, a surprising finding that could be confounded by the exacerbated neuroinflammation seen in this patient and that contrasts with the elevated T cell infiltration reported in active lesions. Thus, the authors would need a comprehensive characterization of MS lesions from prototypical MS specimens to raise such points conclusively. However, given the similarities in neuroinflammation between natalizumab cessation (including JCV seropositivity) and PML, the findings of this study might be more relevant for conditions like PML. This is further supported by the predominant perivascular and parenchymal infiltration of CD8^+^ T cells found in this patient.

Our goal was to use human CNS material with a high level of neuroinflammation and with varying types of lesions in order to evaluate the capacity of IMC to detect numerous markers, simultaneously. As mentioned above, we will refer to this case as an NTZ-rebound case rather than an MS case. We have also stressed in the Discussion the need for further studies on well-characterized cohorts (see Discussion, ninth paragraph).

– On the technical side, the manuscript needs to highlight the benefits of IMC compared to conventional immunostaining. The authors should consider including imaging data highlighting the benefits of IMC such as panels including more than 8 parameters at low and high power will help to appreciate the potential of this technique. Given that most analysis of phenotyping cells were finally made by 3-4 marker combinations it not sufficient elaborated what analyses are done here which are not possible with conventional fluoresecence co-immunostainings.

An important clarification, which admittedly was not made sufficiently clear in the text, is that each region of interest was stained with our entire antibody panel simultaneously. Indeed, the identification of cell phenotypes is based on all markers included in our panel using negative and positive selection. For example, CD8^+^ T cells are identified as being IgM^-^Igκ^-^Igλ^-^ CD45^+^CD3^+^CD8α^+^CD4^-^. Moreover, we limited our depiction of the image so that the reader can view the data in a digestible manner. For example, the images depicting myeloid cells (Figure 4), T cells (Figure 5) and B cells (Figure 6) include a few markers each for easy visualization, and the panels (ROI) shown are exactly the same in each figure. Therefore, for each ROI, we are indeed showing a total of 10-12 markers (across the 3 figures). We have now clarified these points in the text – please see the last paragraph of the subsection “Comparability of IF versus IMC approach and specificity of metal-conjugated antibodies on brain-resident cell types”.

However, we also agree with the reviewer that it is important to highlight the potential of this technique. Thus, we have included a new image that visualizes a combination of 8 markers: Intercalator, PLP, HLA, CD68, CD3, CD8, Igκ/Igλin a single ROI (see Figure 2—figure supplement 1).

– The authors compared only a limited set of markers between IHC and IMC. How did the other markers perform between IHC vs. IMC? In particular, the sensitivity of IHC is strongly dependent on the amplification method used. Thus, stating that IMC might be more sensitive than IHC seems premature at this stage unless supported by a more systematic comparison.

We agree with the reviewer. We have extended the analysis comparing IF and IMC to 3 additional markers, PLP, HLA and CD68. Indeed, we found that the IMC method still reproduces staining patterns that are in agreement with those produced using a standard IF method in MS brain tissue, with similar sensitivity. These data are shown in a new figure (Figure 1—figure supplement 3). We have modified the text to reflect this finding and have removed the comment regarding the relative sensitivity of IMC versus IF since it is impossible to generalize across antibody lots and between IF versus IMC serial sections. These data are now described in the subsection “Comparability of IF versus IMC approach and specificity of metal-conjugated antibodies on brain-resident cell types”.

– When performing the characterization of T cell populations, the authors described a large number of unclassified T cells. The total number of these cells should be included in Figure 7. In future studies, it would be of interest to include markers to determine which cell types are making up such large cohort.

The total number of unclassified T cells is now included in Figure 7. Indeed, it will be of interest to better characterize these cells in the future. We gained some preliminary insight into this with PHATE analysis (see below).

The study would benefit from making use of the novel computational tools available to analyze IMC data. Some examples include PHATE mapping and Pseudotime analysis which would allow to establish phenotypic fates within the immune cell populations studied.

We thank the reviewer for this suggestion and have accordingly performed PHATE analysis in collaboration with Prof David Pitt, who has expertise with this computational method. With his contribution we have been able to include PHATE mapping of brain-resident T cells. The methodology for the PHATE analysis is reported in the subsection “Potential of Heat-diffusion Affinity-based Transition Embedding (PHATE) mapping”; the results are reported in the subsection “Analysis of Potential of Heat-diffusion Affinity-based Transition Embedding (PHATE) mapping” and visualized in Figure 8. Our findings in brief:

1) Using PHATE, CD4^+^ and CD8^+^ T cells are nicely segregated. However, interestingly there is a population CD4^+^, CD8^+^ and unassigned T cells that form a distinct cluster in this particular anti-NTZ-rebound patient. Unassigned T cells are enriched within this distinct cluster. We call this cluster t1.

2) Slowly expanding lesions are the most likely lesion type to contain Cluster t1 cells in this particular NTZ-rebound patient.

3) Based on distance from Collagen^+^ blood vessels, Cluster t1 are almost exclusively found close to blood vessels whereas the CD4^+^ and CD8^+^ T cells that discriminate themselves from Cluster t1 cells are farther from the blood vessels in this particular NTZ-rebound patient.

4) Cluster t1 cells are more likely to have higher levels of CD45 and are less likely to have low levels of HLA (a marker of T cell exhaustion) but seem to not be distinguishable from other lesion-resident cells in terms of Ki67, NFAT or CD38 in this particular NTZ-rebound patient.

– The data analysis workflow as described in this manuscript needs manual intervention at various steps which are not described in sufficient details. This may make it difficult to reproduce the procedure by others in the future.

We agree with the reviewer. We have now included a step-by-step visual description of the manual selection, incorporating examples guiding through the manual classification of myeloid cells, T cells and B cells in Figure 7—figure supplement 2-4.

– Despite the capacity to multiplex stainings for MS, it is not well elaborated what one can learn about the MS pathophysiology by this technology in the current study. The authors should elaborate here more, and this comment relates to the single case study above.

We have elaborated on this in point 1 above and have taken care to not over-interpret our findings for MS in general. Rather, the technique will be quite powerful for larger cohorts of MS patients for which brain specimens are available. Please see the ninth paragraph of the Discussion.

– In the Discussion the authors should consider the state of the literature regarding multiparametric in situ analysis (seminal papers include: Gerner et al., 2012, Moreau et al., 2012, Tsujikawa et al., 2017, Li et al., 2017 and Kwong et al., 2017) and explain how IMC compares to established techniques. As a whole, given the technical nature of the manuscript, the authors should base their conclusions on the experimental underpinnings of IMC and limit extrapolations to MS generally.

We have modified the text to limit conclusions to the single case analysed with no extrapolation to MS in general (see point 1 above). The literature regarding multiparametric in situ analysis (seminal papers include: Gerner et al., 2012, Moreau et al., 2012, Tsujikawa et al., 2017, Li et al., 2017 and Kwong et al., 2017) is now included in the Discussion (ninth paragraph).